# The MURAL collection of prostate cancer patient-derived xenografts enables discovery through preclinical models of uro-oncology

Gail P. Risbridger [1,2,30 ✉], Ashlee K. Clark[1,30], Laura H. Porter[1,30], Roxanne Toivanen[1,2,3], Andrew Bakshi[1,2,3,4], Natalie L. Lister[1], David Pook[1,5,6], Carmel J. Pezaro[1,7,8], Shahneen Sandhu[3,9,10], Shivakumar Keerthikumar[2,3,4], Rosalia Quezada Urban[2,3,4], Melissa Papargiris[1,11], Jenna Kraska[1,2,11], Heather B. Madsen[1,2,11], Hong Wang[1], Michelle G. Richards[1], Birunthi Niranjan[1], Samantha O'Dea[1], Linda Teng[1], William Jenkins[1], Zhuoer Li[12], Nicholas Choo[1], John F. Ouyang [13], Heather Thorne[2,3], Lisa Devereux [2,3], Rodney J. Hicks [14], Shomik Sengupta[7,15,16,17,18], Laurence Harewood[17,19], Mahesh Iddawala[1,20], Arun A. Azad [3,9], Jeremy Goad[3,17,21], Jeremy Grummet[17,22,23], John Kourambas[24], Edmond M. Kwan [5,6], Daniel Moon[17,21,23,25,26], Declan G. Murphy [3,17,21], John Pedersen[1,27], David Clouston[27], Sam Norden[27], Andrew Ryan[27], Luc Furic [1,2,3], David L. Goode[2,3,4], Mark Frydenberg[1,17,23,28,29], Mitchell G. Lawrence [1,2,3,30] & Renea A. Taylor [2,3,12,30 ✉]

Preclinical testing is a crucial step in evaluating cancer therapeutics. We aimed to establish a significant resource of patient-derived xenografts (PDXs) of prostate cancer for rapid and systematic evaluation of candidate therapies. The PDX collection comprises 59 tumors collected from 30 patients between 2012–2020, coinciding with availability of abiraterone and enzalutamide. The PDXs represent the clinico-pathological and genomic spectrum of prostate cancer, from treatment-naïve primary tumors to castration-resistant metastases. Inter- and intra-tumor heterogeneity in adenocarcinoma and neuroendocrine phenotypes is evident from bulk and single-cell RNA sequencing data. Organoids can be cultured from PDXs, providing further capabilities for preclinical studies. Using a 1 x 1 x 1 design, we rapidly identify tumors with exceptional responses to combination treatments. To govern the distribution of PDXs, we formed the Melbourne Urological Research Alliance (MURAL). This PDX collection is a substantial resource, expanding the capacity to test and prioritize effective treatments for prospective clinical trials in prostate cancer.

A full list of author affiliations appears at the end of the paper.

The translation of research discoveries to drug development and ultimately clinical approval is enabled by near-human preclinical models, including patient-derived xenografts (PDXs) and organoids that recapitulate the cancer. However, patient-derived models of prostate cancer are significantly under-represented in international collections such as EurOPDX, Novartis, The Jackson Laboratory (JAX), CrownBio, and the National Cancer Institute's PDXNET and Patient-Derived Models Repository, and rapid screening of promising therapeutic agents in prostate cancer has been significantly hampered by this limitation[1–3].

The therapeutic landscape of prostate cancer has changed dramatically in the last decade, with the introduction of androgen receptor-targeted therapies, including abiraterone acetate and enzalutamide[4], and upfront combination treatments[5,6], which have improved patient outcomes. Most recently, pharmacological inhibitors of poly ADP ribose polymerases (PARP) inhibitors were approved for use in patients with underlying homologous recombination defects, including *BRCA1/2* alterations[7–9]. Other treatments in clinical development include systemic agents and radionucleotide therapies targeting PSMA[10]. However, despite these significant clinical developments prostate cancer remains incurable when men fail these therapies and there is a critical need to identify new treatments to eradicate tumors.

In this study, we present a resource of 59 contemporary PDXs from prostate tumors collected from 2012 to 2020, spanning primary hormone-naïve and metastatic therapy-resistant tumors, including rapid autopsy specimens from men who failed contemporary therapies. This resource provides an opportunity to link the molecular changes in prostate cancer to pathology; grow organoids; and test functional responses to therapies, which have rarely been applied to prostate cancer given the lack of suitable models. Candidate therapies can be tested with a 'one animal per model per treatment' approach ($1 \times 1 \times 1$), an efficient way of screening for active compounds based on striking responses with few biological replicates[1,11], which are then validated with standard preclinical experiments. The resource is available to the research community through the Melbourne Urological Research Alliance (MURAL), and will enable significant advances in the drug development of promising agents.

## Results

### The MURAL prostate cancer PDX collection.
To establish a contemporary collection of prostate cancer PDXs representing a spectrum of aggressive prostate cancer, we collected specimens from men with high-risk localized or metastatic prostate cancer who underwent surgery, biopsy, or rapid autopsy from 2012 to 2020. In total, we established 59 serially transplantable PDXs from 41 tumors of 30 men (Fig. 1). We recorded the pathology of each PDX and the clinical data of each patient. Multiple PDXs were established from some patients, including matching primary tumors and metastases (167.1R & 167.2M and 452C & 426M), and matching metastases collected from different sites via rapid autopsy (27A, 201A, 435A, and 463A). All PDXs were established in mice supplemented with testosterone. Similar to previous studies[12–17], we also grew 18 PDXs as sublines in castrated mice (PDX-Cx), where androgen levels approximate patients treated with abiraterone[18].

This cohort of PDXs was established from patients across the disease trajectory of prostate cancer from treatment-naïve to metastatic castrate-resistant disease, including men who received conventional and experimental systemic treatments (Supplementary Data 1). The PDXs of primary prostate tumors were grown from radical prostatectomy samples of treatment-naïve patients, except for 452C, which was from a treated tumor obtained via

transurethral resection of the prostate (TURP) (Supplementary Data 1). Most metastases were from patients who had progressed on androgen deprivation therapy (ADT), except the samples from patients 395 and 426. Of the patients with metastatic CRPC, all but three (373, 374, and 424) had also received other systemic treatments, including AR-directed therapies (abiraterone and/or enzalutamide), chemotherapy (docetaxel, cabazitaxel, carboplatin), or novel agents ([77]Lu-PSMA, PARP inhibitor (niraparib), immunotherapy (pembrolizumab)) (Fig. 1 and Supplementary Data 1). Some patients had progressed on multiple lines of treatment after ADT, particularly those who had consented to rapid autopsy. Twenty patients have died from prostate cancer (including the 6 patients who had rapid autopsies; Fig. 1).

The expression of common prostate cancer biomarkers, including AR, AMACR, PSA, PSMA, ERG, and neuroendocrine markers (synaptophysin, chromogranin A and CD56) were continually assessed across PDX generations. In addition, the identity of PDXs was routinely authenticated by profiling short tandem repeats (STRs). Immunostaining for CD45 was used to rule out the presence of lymphoma, which has contaminated PDXs in the previous studies[19,20] (see "Methods"). Figure 1 shows data from the latest PDX generation (refer to Supplementary Data 1 for the latest generation of each PDX), and demonstrates their diverse phenotypes. Based on histopathology, there are three main groups of PDXs: adenocarcinomas (55%), neuroendocrine tumors (31%), and those with mixed pathology that are positive for both the AR and neuroendocrine markers to varying degrees (14%; Fig. 1). Among the adenocarcinomas, all PDXs are AR+, except 201.2A and 201.2A-Cx, which lack the AR but express CD56[21]. ERG (24% positive) and PSMA (64% positive) staining also varied across the cohort (Fig. 1). Biomarker expression in the PDXs was consistent with expression in the original patient specimens (Supplementary Data 2). Overall, this collection of PDXs spans the clinical heterogeneity of prostate tumors from diagnosis to death, including a range of adenocarcinoma (AR-positive), neuroendocrine (AR-null), and mixed phenotypes.

### Pathological and clinical features of tumors grown as PDXs.
Prostate cancer is considered difficult to grow as PDXs, so we examined the features underpinning the success of this collection. Overall, 208 specimens from 88 prostate cancer patients were grafted into immune-deficient mice (Supplementary Data 3, Supplementary Fig. 1a). Of the 63 primary tumor samples obtained from radical prostatectomy or TURP, 39 (61.9%) survived for at least one generation, and 13 (20.6%) formed serially transplanted PDXs (Fig. 2a, Supplementary Data 3). We compared the serially transplantable and non-serially transplantable primary tumors. There was no significant difference in the percentage of Ki67 cells in the original patient specimens, the time to the first PDX generation, or grade group; however, serially transplantable PDXs were from significantly larger volume tumors (Fig. 2b–e). Moreover, the serially transplantable primary tumors were from patients with significantly poorer overall survival from prostate cancer (Fig. 2f; HR 10.93 (95% CI = 1.51–79.08). Thus, clinically aggressive primary prostate cancers appear more likely to grow as serially transplantable PDXs, which are a group especially in need of novel therapeutics.

We also grafted 145 metastases from 38 men (Fig. 2g, Supplementary Data 3). Of these samples, 29 (20%) survived as PDXs for at least one generation, and 28 (19.3%) were serially transplantable (Fig. 2g, Supplementary Data 3). The serially transplantable PDXs were from metastases to lymph nodes and various soft tissues, including brain, liver, and lung. Unlike the primary tumors, serially transplantable metastases were from samples with significantly higher Ki67 staining (Fig. 2h), and had a shorter time from when the patient tissue was initially

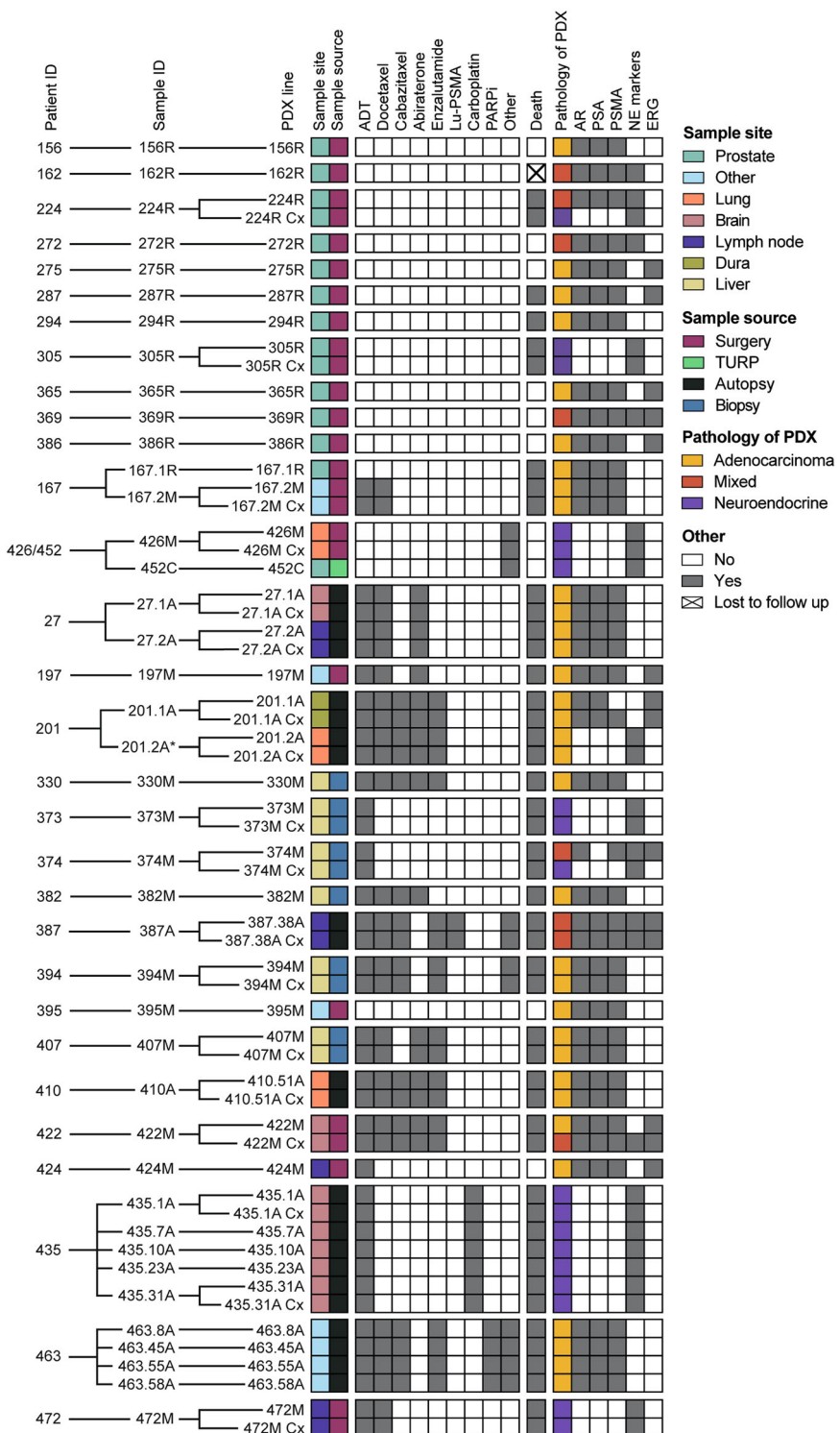

**Fig. 1 MURAL Prostate Cancer PDX collection.** Heatmap summarising 59 prostate cancer PDXs established from 41 specimens obtained from 30 patients. The sample site, sample source, systemic therapies administered to patients prior to sample collection, clinical outcome at last follow-up, and pathology and biomarker expression of the PDXs are shown. Pathology of the PDXs was determined through histology review by pathologists and expression of phenotypic biomarkers by immunohistochemistry. NE marker staining indicates expression of ≥1 of chromogranin A, synaptophysin, and CD56. Immunohistochemistry results are from the latest PDX generation. *PDX 201.2A and PDX 201.2A-Cx are classified as adenocarcinoma based on pathology review and negative staining for synaptophysin and chromogranin A; however, they have focal staining of CD56. The naming convention for PDXs is a follows: numbers indicate patient ID and tumor site (e.g., 167.1 is patient 167, site 1), letters denote the sample source (R—radical prostatectomy; M—biopsy or surgical sample of metastasis; C— castration-resistant primary tissue; A—autopsy tissue), and Cx denotes subline grown in castrated mice.

implanted to the first generation it was harvested as a PDX (Fig. 2i). The take rate of PDXs also varied based on the site and source of tissue. For example, brain metastases had a high take rate (58%), but bone metastases did not (0%) (Supplementary Data 3, Supplementary Fig. 1a). Notably, the take rate was significantly higher for samples from biopsy or surgery (84%) versus rapid autopsy (11%) (Supplementary Data 3, Supplementary Fig. 1a). This is likely due to tissue degradation related to the additional time it took to collect rapid autopsy samples, with a median of 6.8 h (range 4–10.5 h) between death and the start of autopsy (Supplementary Fig. 1b), whereas biopsies and surgical samples were transferred directly to the laboratory. Based on AR staining of 110 metastases, AR-positive tumors (including mixed tumors) and AR-negative tumors had an equivalent take rate (Fig. 2j).

**Divergent transcriptomic profiles of AR-responsive and NE phenotypes within and between PDXs.** To examine the spectrum of tumor phenotypes in the MURAL cohort, we performed RNA-seq on 89 samples from 39 PDXs from testosterone-supplemented or castrated mice. The cohort clustered into two main groups across principal component 1 (19.9% variation): AR-positive PDXs, and neuroendocrine PDXs. Tumors with mixed pathology were at the border between groups in most cases (Fig. 3a). Hierarchical clustering of single-sample gene set enrichment analyses for hallmark pathways also largely separated PDXs into AR-positive and neuroendocrine groups (Supplementary Fig. 2a). Confirming this observation, well-characterized transcriptional signatures of the androgen response[22] and the neuroendocrine phenotype[23] also separated AR-positive and neuroendocrine PDXs into different clusters, with the mixed tumors in between each cluster (Fig. 3b). Thus, based on their transcriptome profiles, the PDXs primarily clustered by pathology, rather than stage of progression (primary vs. metastatic) or growth in testosterone-supplemented versus castrate host mice.

To further examine the PDXs with mixed pathology, which had an intermediate phenotype in bulk RNA-seq analyses, we used histopathology and single-cell RNA-seq (scRNA-seq) to examine the subpopulations of cells within them. Histopathology showed that PDX 224R has mixed pathology, with clusters of AR-positive cells intermingled with NCAM1 (CD56)-positive cells (Fig. 3c; also see Fig. 4b). UMAP analysis of scRNA-seq separated PDX 224R into 3 adenocarcinoma clusters and 5 neuroendocrine clusters (Fig. 3d), which varied in AR and NCAM1 (CD56) expression (Fig. 3e), and enrichment of NE and AR signatures (Fig. 3f). A proliferative signature was enriched in the adenocarcinoma AD3 and neuroendocrine N1 clusters, while an EMT signature was enriched in two neuroendocrine clusters (N4 and N5; Fig. 3f).

Since scRNA-seq identified clusters within adenocarcinoma, we also used it to examine PDX 287R, which has homogenous adenocarcinoma pathology and AR expression, and no staining of neuroendocrine markers (Fig. 3g; also see Fig. 4b). The UMAP for PDX287R also had multiple clusters of adenocarcinoma cells, with varying enrichment of AR and proliferative signatures (Fig. 3h–j). Thus, based on bulk RNA-seq and scRNA-seq, the MURAL PDX collection encompasses diverse tumor phenotypes, even within individual PDXs.

**Somatic mutation frequency and genomic features of PDXs match the clinical landscape.** We used targeted DNA sequencing to determine the genomic features of each PDX, focussing on copy number variations and mutations at high allele frequency in a curated set of alterations previously identified in large patient cohorts of prostate cancer (Supplementary Data 4)[24,25]. See Supplementary Fig. 3 for genomic alterations across the MURAL PDX collection and Supplementary Data 5–6 for functional variants. The percentage of genome alteration (PGA) varied between PDXs, but was higher on average in PDXs of metastases versus those from primary tumors, consistent with patient cohorts (Fig. 3k)[24].

The four most common somatic alterations in MURAL PDXs were deletions or mutations of TP53, PTEN, and RB1, and amplifications of MYC (Fig. 3l; gene list per PDX is provided in Supplementary Fig. 3a). This reflects that MURAL PDXs were primarily established from high-risk primary tumors and metastases, since these prominent genomic drivers of prostate cancer are enriched in metastases versus primary tumors in patient cohorts[24]. One PDX had a pathogenic germline alteration, with loss of BRCA2 in PDX 294R (Supplementary Fig. 3a). The "long tail" of other alterations included genes in the AR, DNA damage repair, PI3 kinase, and Wnt pathways (Fig. 3l, Supplementary Fig. 3b). As expected, alterations of the AR were common in PDXs of castrate-resistant adenocarcinoma, including amplifications, mutations, and structural rearrangements (Supplementary Fig. 3b). DNA damage repair defects included loss of BRCA2 (PDXs 294R, 435.1A, 435.7A, 435.10A and. 435.23A), MLH1 (PDXs 201.1A, 201.1A-Cx), and MSH2 (PDXs 272 R, 287 R) (Supplementary Fig. 3b). Among the alterations in the Wnt pathway, there was the loss of APC in PDXs from two patients (PDXs 167.2M, 167.2M-Cx, 407M, 407M-Cx) (Supplementary Fig. 3b). In the PI3 kinase pathway, there were frequent deep deletions of PTEN as well as amplifications of PIK3CA and/or PIK3CB (Supplementary Fig. 3b). Thus, these genomic features are representative of the genomic spectrum observed in clinical specimens.

Further comparison of the genomic profiles of MURAL PDXs showed that later generation castrate sublines largely maintained the genomic features of matching, earlier generation PDXs grown in testosterone-supplemented mice (Supplementary Fig. 3c). There was minimal change in the allele frequency of functional variants in most PDXs, except for PDX 201.1A versus PDX 201.1A-Cx, where several new variants arose over time, likely driven by the mismatch repair defect in this tumor (Supplementary Fig. 3b, c). After castration, there was a trend of increased AR copies in two PDXs (167.2M-Cx and 394M-Cx), but little change in the other PDXs, and no new AR mutations (Supplementary Fig. 3d, Supplementary Data 5–6).

We also noted expected differences in the frequency of alterations based on pathology and stage of disease progression. PDXs with NE pathology tended to have more frequent alterations in TP53, PTEN, and MYC, but less frequent alterations in AR, compared to PDXs with adenocarcinoma pathology (Supplementary Fig. 3e). Similarly, alterations of RB1, APC, and AR were more common in PDXs from metastases versus those from primary tumors (Supplementary Fig. 3e).

**PDX tissue can be grown as organoids.** Xenografts and organoids are complementary models for preclinical testing. Therefore, we examined whether cells from 24 MURAL PDXs could be grown as organoids, also known as tumoroids. Using previously described protocols[26,27], 22 PDXs (92%) grew for at least one passage (Fig. 3m). Five tumors (21%) displayed active growth, with increasing population doublings over several passages, including primary tumors and metastases with adenocarcinoma or neuroendocrine pathology (Fig. 3m, n, Supplementary Fig. 2b and Supplementary Data 7). The other 17 tumors that established as organoids had limited

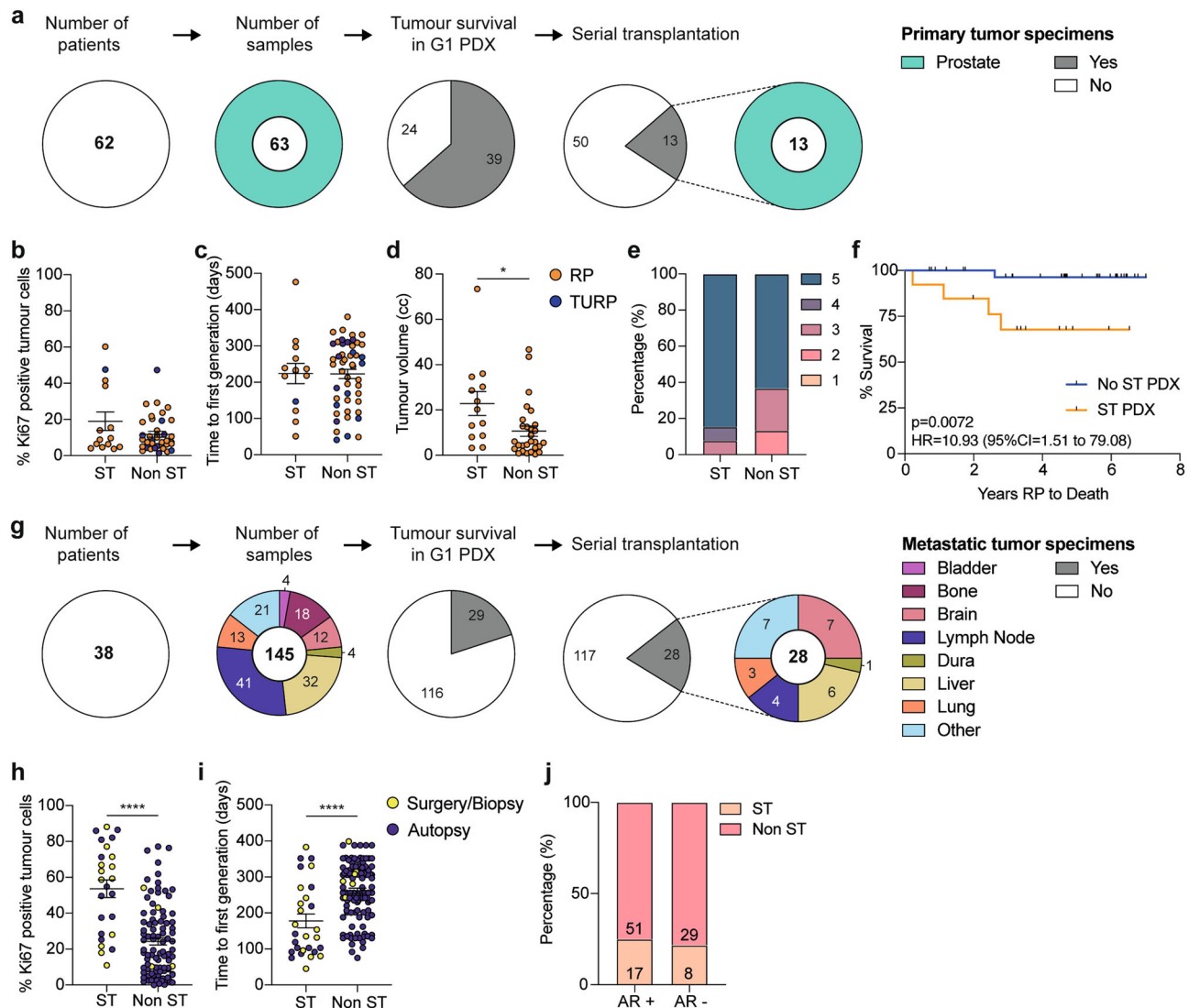

**Fig. 2 Pathological and clinical features of tumors used to establish PDXs. a** Pie charts show the number of patients consented and the number of primary prostate tumor samples collected for xenografting compared to the number of samples that maintained tumor tissue in the first generation (G1) PDX and established as serially transplantable (ST) PDXs. Color denotes the site that each sample was taken from. **b–d** The percentage of Ki67-positive tumor cells in pre-grafted tissue (**b**; $n = 14$ ST, 38 non ST), time to first generation (**c**; $n = 14$ ST, 49 non ST), and tumor volume at surgery for radical prostatectomy (RP; orange) and transurethral resection of the prostate (TURP; purple) specimens (**d**; $n = 13$ ST, 28 non ST, $P = 0.016$) that established ST PDXs compared to those that did not. Unpaired two-sided $T$ test for ST vs non ST; data shown as mean ± SEM. **e** The percent of primary tumors with a Gleason grade group of 1–5 that did ($n = 13$) or did not establish ST PDXs ($n = 30$; not significant, Mann Whitney test comparing the distribution of Gleason grade groups between ST vs non ST). **f** Kaplan–Meier curve comparing the survival of patients whose RP specimen did (orange; $n = 13$) or did not (blue; $n = 37$) establish ST PDXs. $P = 0.0072$; log rank test; HR = 10.93; 95% CI 1.51 to 79.08. **g** Pie charts show the number of patients consented and the number of metastatic tumor samples collected for xenografting compared to the number of samples that maintained tumor tissue in the G1 PDX and established as serially transplantable PDXs. Color denotes the site that each sample was taken from. **h–i** The percentage of Ki67-positive tumor cells in pre-grafted tissue (**h**; $n = 25$ ST, 94 non ST, $P < 0.0001$), and time to first generation (**i**; $n = 28$ ST, 117 non ST, $P < 0.0001$) for metastatic tumor samples obtained from surgery/biopsy (yellow) or autopsy (purple) that did or did not establish ST PDXs. Unpaired two-sided T test for ST vs non ST; data shown as mean ± SEM. **j** The percentage of androgen receptor (AR)-positive and AR-negative metastatic tumors that did ($n = 25$) or did not establish ST PDXs ($n = 80$), based on immunohistochemistry for AR in the original tumor tissue. Not significant; Fisher's exact test. Source data are provided as a Source Data file.

growth, and could not be maintained for more than three passages in culture (Fig. 3m and Supplementary Data 7). Organoids had the same expression profile of AR and neuroendocrine markers as the original PDXs (Fig. 3o). They also clustered with their corresponding PDXs in analysis of bulk RNA-seq data and hierarchical clustering of AR and NE signatures (Supplementary Fig. 2c, d). Thus, PDX tissues can be grown as organoids and retain tumor phenotypes, providing greater flexibility for preclinical testing.

**Research-ready serially transplantable PDXs for preclinical testing.** To prioritize a subset of PDXs for routine preclinical testing, we selected 17 research-ready PDXs (Fig. 4a). PDXs were classified as research-ready if they have rapid and consistent turnover in host mice, and are able to grow subcutaneously to allow for continual tumor measurements. Supplementary figures detail the clinical history of each patient (Supplementary Fig. 4); the response of each PDX to castration (Supplementary Fig. 5a); the growth rates and transplantation of each PDX over time in

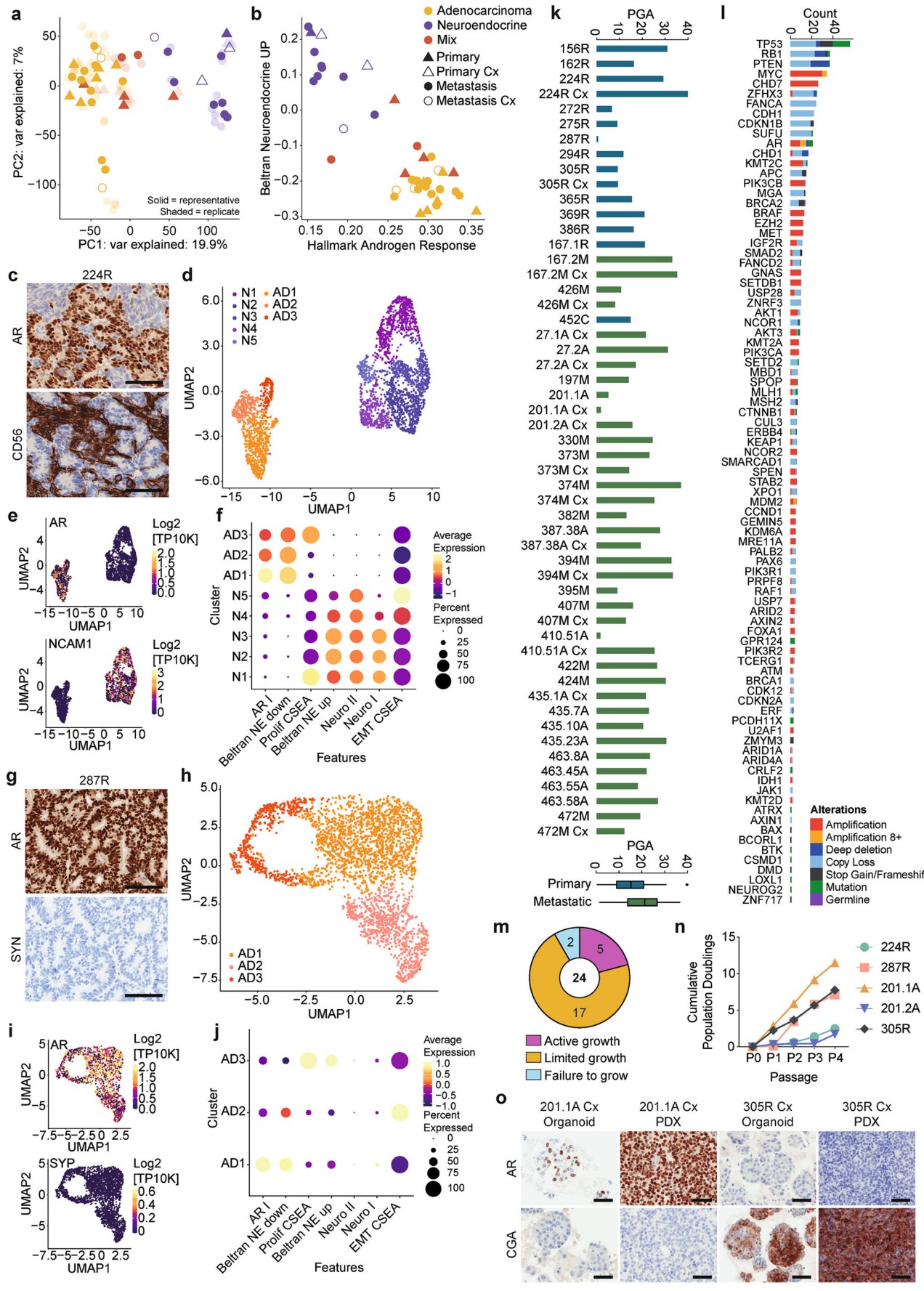

**Fig. 3 Phenotypic and genomic heterogeneity within and between PDXs. a**, **b** Principle component (PC) analysis of gene expression from RNA sequencing in PDXs grown in testosterone-supplemented (+T; filled symbol) or castrate (Cx; empty symbol) mice. PDXs from metastases are represented by circles, and primary tumors by triangles. PDX pathology is indicated (adenocarcinoma—yellow; neuroendocrine—purple; mixed— red). Representative samples from each PDX are colored symbols (PDXs from primary prostate cancer—triangles, PDXs from metastatic prostate cancer—circles), while replicates are shown as transparent symbols. **a** PCA plot based on PC1 and PC2, showing the two largest sources of variation in the expression of genes across PDXs. **b** Plot of gene set enrichment analysis using Singscore to compare Beltran Neuroendocrine Score to Hallmark Androgen Response Score in PDXs. **c**–**j** Single-cell RNA sequencing reveals intra-tumoral heterogeneity of adenocarcinoma (AD) and neuroendocrine (NE) PDXs. **c** Representative immunohistochemical staining of PDX 224R for androgen receptor (AR) and NE marker NCAM1 (CD56; scale bars = 100 μM). **d**–**e** UMAP (**d**) and marker gene expression (**e**) of PDX 224R showing the presence of 3 AD and 5 NE clusters based on single-cell RNA sequencing. **f** Dot plot comparing expression of AR[79], NE gene signatures (Beltran NE[80], Neuro I, Neuro II[79]), and hallmark signatures (proliferation and epithelial-mesenchymal transition: EMT[81]) across the AD and NE clusters for PDX 224R. **g** Representative immunohistochemical staining of PDX 287R for AR and synaptophysin (SYN; scale bars = 100 μM). **h**–**i** UMAP (**h**) and marker gene expression (**i**) of PDX 287R showing the presence of 3 AD clusters. **j** Dot plot comparing expression of AR[79], NE gene signatures (Beltran NE[80], NeuroI, NeuroII[79]), and hallmark signatures (proliferation and epithelial-mesenchymal transition: EMT[81] across the AD clusters for PDX 287R. **k**–**l** Somatic mutation frequency and genomic-landscape analyses based on targeted DNA sequencing. **k** The percent genome alteration (PGA) in PDXs from testosterone-supplemented and castrated (Cx) host mice from primary (blue) and metastatic (green) samples. **l** The number of somatic alterations per gene across the PDX cohort. Somatic nucleotide variations with 0.75 or greater allelic frequency are reported (amplification—red; amplification with 8 or more copies—orange; deep deletion—dark blue; copy number loss—light blue (i.e. fewer copies than baseline ploidy); stop gains and frameshift mutations—black; missense mutation—green; germline mutation—purple). **m**–**o** Phenotypic heterogeneity is maintained in organoids established from PDXs. **m** Pie chart showing the growth of 24 PDXs as organoids (active: increased population doublings ≥4 passages (purple); limited: growth/survival for ≤3 passages (orange); failure: poor growth (blue)). **n** Cumulative population doublings population doublings of organoids organoids across passages. Source data are provided as a Source Data file. **o** Representative immunohistochemical staining for AR and chromogranin A (CgA) in PDX tissue and organoids from PDX 201.1A-Cx and PDX 305R-Cx (scale bars = 50 μM). For panels with NE markers, the highest expressed marker is shown. Staining is repeated every fifth generation across all PDXs, and for representative organoid cultures.

testosterone-supplemented (Supplementary Fig. 5b) and castrated mice (Supplementary Fig. 5c); and the histopathology of each PDX (Supplementary Fig. 5d).

Like the full MURAL cohort, the research-ready PDXs represent tumors with diverse clinical trajectories, pathologies, and genomic features. This subset of tumors includes 4 PDXs from testosterone-supplemented mice, and 13 PDXs from castrated mice, including two PDXs of primary tumors (PDX 224-Cx and PDX 305-Cx; Fig. 4a). Pathology review and immunohistochemistry for phenotypic markers (AR, PSA, PSMA, ERG, chromogranin A, synaptophysin, and CD56) showed that 9 research-ready PDXs are adenocarcinoma, 7 are neuroendocrine, and 1 has mixed pathology (Fig. 4b). Of the 9 AR-positive PDXs, 5 express AR-V7 and/or ARv567es, including PDX 27.1A, which lacks full-length AR expression as previously reported[21] (Fig. 4b). The research-ready PDXs have genomic alterations commonly reported in patient cohorts, based on targeted DNA sequencing, and analysis of variants present at high allele frequency (Fig. 4c). Every research-ready PDX has a copy number loss, deep deletion, or mutation of either *TP53*, *RB1*, or *PTEN*, and 12 PDXs have alterations of at least two of these tumor suppressor genes. Alterations in AR, Wnt, DNA damage repair, and PI3 Kinase pathways are also represented in the research-ready PDXs.

As case studies, PDX 287R and PDX 435.1A-Cx exemplify the diverse features of the research-ready PDXs. PDX 287R was established from a man with treatment-naïve primary prostate cancer who donated radical prostatectomy tissue. The patient subsequently had salvage radiation therapy and ADT, but died 3 years after diagnosis without any other systemic treatments (Fig. 4d). PDX 287R was established in testosterone-supplemented mice and, consistent with its origin as a castrate-sensitive tumor, regressed in castrated mice (Fig. 4e). PDX 287R was initially grafted in 2014, and has an average generation time of 57 days (Fig. 4f, Supplementary Data 1). It expresses AR, PSA, PSMA, and ERG, but not neuroendocrine markers (Fig. 4b and Supplementary Fig. 5d). In contrast, PDX 435.1A-Cx is from a brain metastasis of a patient with CRPC who consented to rapid autopsy (Fig. 4g). He died 8 years after diagnosis, after two rounds of radiotherapy, 5 years of ADT, the emergence of neuroendocrine pathology, and treatment with carboplatin and

etoposide (Fig. 4g). PDX 435.31A was established in testosterone-supplemented mice, and continued to grow as PDX 435.1A-Cx in castrated mice, consistent with its origin as a CRPC (Fig. 4h). PDX 435.1A-Cx was initially grafted in 2018, and has an average generation time of 56 days (Fig. 4i). This PDX did not express AR, PSA, PSMA or ERG, but was positive for CD56, synaptophysin and low levels of chromogranin A (Fig. 4b and Supplementary Fig. 5d). Collectively, the research-ready PDXs have heterogenous features, providing diverse tumors for preclinical drug testing.

**Preclinical testing of combination therapies in research-ready PDXs.** PARP inhibitors have recently been approved as monotherapy for patients with advanced prostate cancer with defects in homologous recombination who have failed a second-generation hormonal agent (and a taxane chemotherapy in the case of rucaparib)[7,28]. We used the research-ready PDXs to examine whether combining PARP inhibitors with other treatments could also inhibit the growth of castration-resistant tumors lacking genomic DNA repair defects. Ongoing clinical trials are combining PARP inhibitors with chemotherapies, AR-directed treatments, and other DNA damage-inducing agents[29–31]. To represent these different approaches, we selected apalutamide, enzalutamide, azacitidine (hypomethylating agent), AZD1775 (Wee1 inhibitor), VX-970 (ATR inhibitor), docetaxel, and carboplatin for combination therapies. We performed these studies using talazoparib, a highly potent PARP inhibitor that induces DNA damage at low concentrations by inhibiting PARP1/2 catalytic activity and trapping PARP at single-strand DNA breaks[32,33]. Talazoparib has FDA approval for HER2-negative breast cancer with germline *BRCA1/2* mutations, and is currently in clinical trials for metastatic prostate cancer, alone and in combination with other agents[29].

We used the one animal per model per treatment approach (1 × 1 × 1), developed by Migliardi and team[11] and validated at scale by Gao and colleagues[1], to rapidly screen 8 research-ready PDXs with seven different combination treatments (Fig. 5a). As a positive control for PARP sensitivity, we used PDX 435.1A Cx, which has a bi-allelic *BRCA2* deletion (Fig. 4f). We also selected four AR-positive (PDXs 27.1A-Cx, 27.2A-Cx, 167.2M-Cx, and

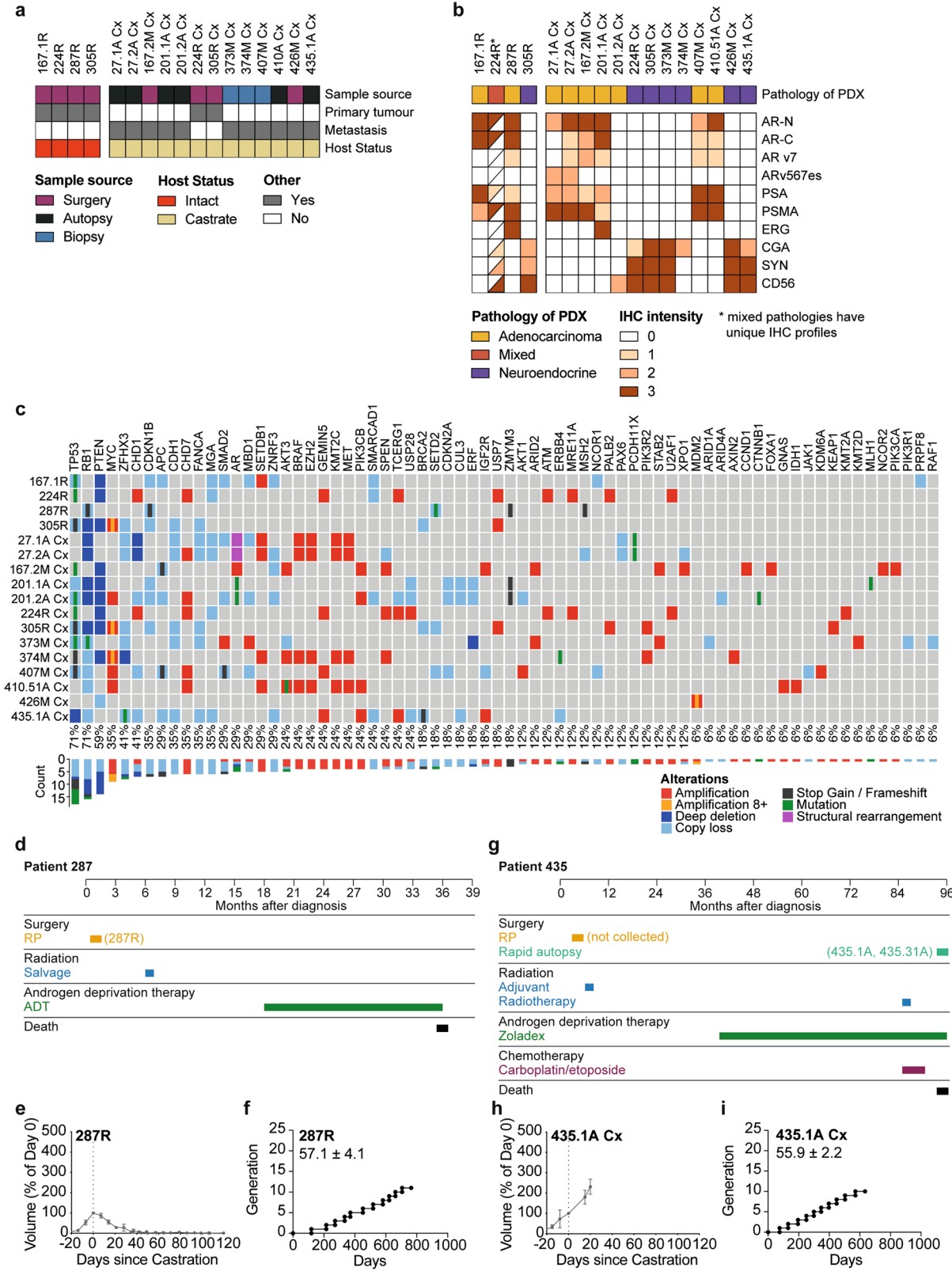

201.1A-Cx) and three AR-null (PDXs 201.2A-Cx, 373M-Cx, 224R-Cx) research-ready PDXs lacking homozygous alterations in DNA repair genes (Fig. 4d). Based on the growth of our PDXs over 4 weeks, we pre-defined the thresholds for responder (<100% growth), partial responder (100–300% growth and ≤50% volume of matched vehicle), and non-responder (>300% growth), based on changes in graft volume from the start of treatment.

Varying responses to the combination therapies were observed across the PDXs, with the most effective combinations being talazoparib + carboplatin and talazoparib + docetaxel as 6/8 tumors had a response or partial response (Fig. 5a, Supplementary Fig. 6a). Representative graphs show the striking tumor regression of PDX 435.1A-Cx treated with talazoparib + carboplatin compared to the vehicle control, as expected given the loss

**Fig. 4 Applying serially transplantable PDXs as 'research-ready' models for preclinical testing. a** Heatmap shows sample source (surgery—purple; autopsy—black; biopsy—blue), tumor type (indicated as yes (gray) or no (white)), host mouse status (intact—red; castrate— yellow) of 17 PDXs that have been classified as 'research-ready' for preclinical testing based on rapid turnover and subcutaneous growth in host mice. **b** Heatmap shows pathology (adenocarcinoma—yellow; neuroendocrine—purple; mixed—red) and immunohistochemistry (IHC) analysis of the research-ready PDXs (staining intensity indicated as 0, 1, 2, 3 and depicted as a gradient of brown coloring). Half-filled squares indicate mixed IHC staining in PDX 224R due to mixed adenocarcinoma and neuroendocrine pathology. **c** Summary of key somatic alterations in research-ready PDXs based on targeted DNA sequencing. The percent of PDXs with alterations in individual genes are shown. The bar plot shows the number of alterations observed in individual genes across the PDX cohort. Somatic nucleotide variations with 0.75 or greater allelic frequency are reported (amplification with three or more copies—red; amplification with eight or more copies—yellow; deep deletion—dark blue; copy loss—light blue; stop gains and frameshift mutations—black; missense mutation—green; structural rearrangements—pink). **d** Treatment timeline from diagnosis to death for patient 287. **e** Tumor volume, presented as a percent change in tumor volume from castration (day 0; $n = 6$ grafts) and **f** growth trajectory of PDX 287R. The average time per generation is 57.1 ± 4.1 days for PDX 287R in testosterone-supplemented host mice. **g** Treatment timeline from diagnosis to death for patient 435. **h** Tumor volume, presented as a percent change in tumor volume from castration (day 0; $n = 3$ grafts) and **i** growth trajectory of PDX 435.1A-Cx. The average time per generation is 55.9 ± 2.2 days for PDX 435.1A-Cx in castrate host mice. **e, h** Grafts were established subcutaneously in testosterone-supplemented mice until tumor volume reached 200 mm³, at which point host mice were castrated ($n = 2$–4 grafts; data shown as mean ± SEM). **f, i** Each data point represents a different generation with a ≥10-fold increase in tumor volume.

of *BRCA2* (Fig. 5b, Supplementary Fig 6b). Interestingly, PDX 224R-Cx also regressed, and four other PDXs had partial responses (Fig. 5b).

To further investigate the efficacy of talazoparib + carboplatin, we performed expansion studies with three PDXs that had varying responses in the 1 × 1 × 1 experiments; PDX 224R-Cx, a responder, and PDXs 201.1A-Cx and 167.2M-Cx, both partial responders. We also selected two previously untested PDXs, 305R-Cx which has mono-allelic *BRCA2* deletion, and 426M-Cx, which has no known genomic defects in DNA damage repair (Supplementary Fig. 3b). These PDXs represent the phenotypic heterogeneity within the MURAL cohort, with both AR-positive and AR-null neuroendocrine pathology. PDXs were treated with vehicle, single-agent (talazoparib or carboplatin), and combination treatment ($n = 6$–8 per group). We measured the average growth rate in each group using a mixed model analysis (Supplementary Data 8), the tumor volume of individual mice, the change in tumor volume across the whole treatment period, and the length of time on treatment for each PDX (Fig. 5c–e). Collectively, these analyses showed that combination treatment was most effective, significantly decreasing tumor growth rate compared to vehicle in 4/5 of the PDXs.

The only PDX that did not decrease in average tumor volume following combination therapy was PDX 201.1A-Cx. Most grafts of this rapidly growing PDX reached the maximum ethical volume before the end of the 28-day treatment period, thus reducing treatment time (Fig. 5d, Supplementary Fig. 6c). Some individual grafts in the treatment groups had slower growth rates, similar to the 1 × 1 × 1 experiments with this PDX (Fig. 5d). Unlike PDX 201.1A-Cx, PDX 224R-Cx was highly sensitive to the combination therapy, confirming the results of the 1 × 1 × 1 study. The combination treatment and talazoparib alone both caused tumor regression of PDX 224R-Cx, based on reduced tumor volume compared to day one of treatment (Fig. 5c–e). Moreover, talazoparib, carboplatin, and the combination treatment all significantly decreased average tumor volume compared to vehicle, despite the lack of genomic defects in the DNA damage repair pathway in this tumor (Fig. 5c–e).

The other three PDXs had partial responses to talazoparib + carboplatin therapy, with significant reductions in average tumor volume and delayed tumor growth compared to vehicle, but no tumor regression (Fig. 5c–e). Notably, PDX 305R-Cx, which has *BRCA2* copy number loss, had reduced tumor volume with talazoparib treatment alone compared to vehicle control, consistent with occasional responses to PARP inhibitors in patients with monoallelic *BRCA2* loss (Fig. 5c–e)[34]. PDX 426M-

Cx was sensitive to carboplatin treatment alone (Fig. 5c–e). The combination of talazoparib + carboplatin was well tolerated, with minimal change in mouse body weights over the 28-day treatment period (Supplementary Fig. 6d). Complete data for individual PDX experiments is provided in Supplementary Fig. 6a–d. These data demonstrate the usefulness of research-ready PDXs to validate drug combinations.

**Distributing the MURAL PDX collection.** MURAL PDXs are enduring resources that can be shared with other academic investigators or pharmaceutical companies. To facilitate their distribution, we established the MURAL consortium in 2017 (Fig. 6), as a collaboration between researchers, clinicians, and patient advocates at Monash University and Peter MacCallum Cancer Centre. Our goal was to create a PDX biospecimen resource for collaborative, investigator-led research for use in peer-reviewed, ethically approved, and funded research projects in urology and oncology[35].

MURAL currently has 12 approved projects, and in 2019–20, distributed >30 PDX or organoid samples to nine researchers (three Australian-based and six international). MURAL provides access to fresh, fixed, and frozen samples of PDX tissue; cryopreserved PDXs for re-implanting into mice; cryopreserved organoid cultures; DNA and RNA profiles; scRNA-seq data; immunohistochemistry data for biomarker expression; mouse serum for investigating circulating biomarkers; and clinical data from patients who donated tumors. MURAL requires researchers to provide regular updates and reports to minimize duplication and ensure meaningful outcomes are produced.

## Discussion

PDX studies have become a crucial step in cancer drug development. The effectiveness of the 1 × 1 × 1 experimental paradigm was demonstrated for multiple tumor types using the Novartis Institutes for Biomedical Research patient-derived tumor xenograft encyclopedia (NIBR PDXE)[1], but prostate cancer was not included among ~1000 PDXs. This under-representation of prostate cancer in large repositories of patient-derived models could lead to prostate cancer patients being excluded from phase 1 clinical trials of new drugs due to a lack of preclinical data. Here we report a collection of 59 prostate cancer PDXs, including 17 research-ready lines for preclinical drug testing. We document the success rate of establishing serially transplantable PDXs, and report the clinical, pathological, and genomic features of heterogeneous prostate tumors. We also demonstrate their utility for rapidly screening single agent and combination treatments to

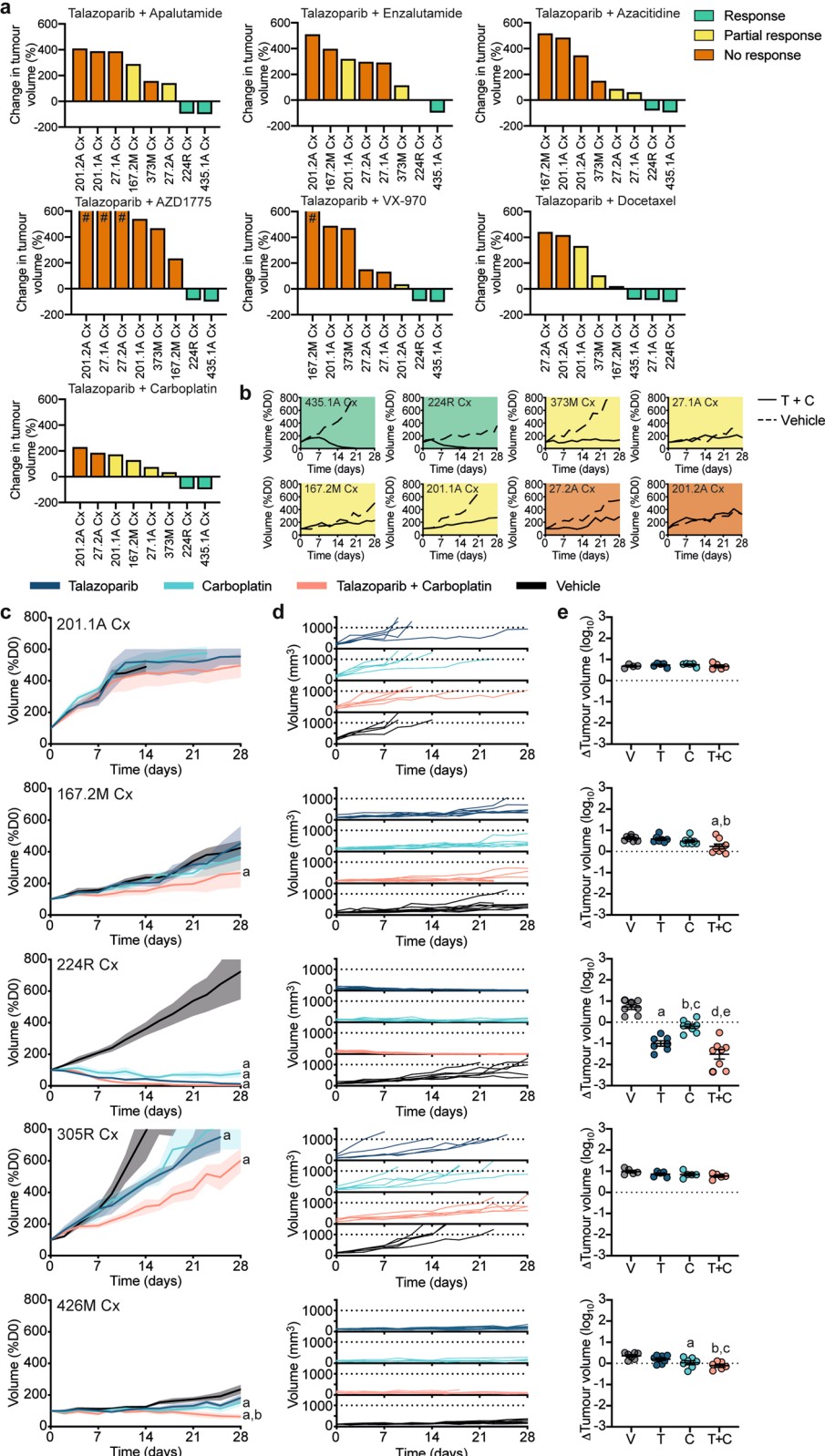

identify exceptional responders. These PDXs provide a strong platform for preclinical testing that may prompt clinical trials of therapies for advanced prostate cancer.

Prostate cancer is one of the most commonly diagnosed malignancies, but it has long been difficult to grow as PDXs or organoids, with low take rates and long latency times[26,36–38]. Several laboratories have devoted expertise and resources into

generating these models, and an international consortium of 11 laboratories, including our own, recently described 98 PDXs of prostate cancer[39]. This collection includes significant cohorts of PDXs from the LuCAP[12], MD Anderson (MDA PCa PDX)[14,40,41], Rotterdam[17,42], Living Tumor Laboratory (LTL)[13,43] and Hopkins[44,45] PDX collections. Our cohort of 59 PDXs builds upon this consortium, and other collections[46],

**Fig. 5 Preclinical testing of combination therapies in serially transplantable PDXs. a** Waterfall plots of the response of eight research-ready PDXs to talazoparib combination therapies using the one animal per model per treatment (1 × 1 × 1) approach after up to 28 days of treatment. Data presented as the percent change in tumor volume compared to day 0 of treatment (%D0), with a good response shown in green (tumor volume regressed to <100% of starting volume), a partial response shown in yellow (tumor volume between 100–300% of starting volume and ≤50% volume of matched vehicle) and no response shown in orange (tumor volume >300% of starting volume), # tumor volume increases over 600% are not represented. (**b**) Graphs show tumor volume (%D0) for PDXs treated with vehicle (dotted line) or talazoparib (T) and carboplatin (C) combination therapy (solid line) for up to 28 days using the 1 × 1 × 1 approach (response—green; partial response—yellow; no response— orange). **c–e** Expansion of talazoparib and carboplatin combination therapy in five PDXs. Mice were treated for up to 28 days with vehicle (V; black; $n = 6$–8 grafts), 0.33 mg/kg talazoparib (T; dark blue; $n = 6$–8 grafts), 50 mg/kg carboplatin (C; light blue; $n = 6$–8 grafts) or talazoparib and carboplatin (T + C; pink; $n = 6$–8 grafts). Graphs show (**c**) tumor volume (%D0) for treatment groups (mean ± SEM; [a]$P < 0.05$ compared to vehicle, [b]$P < 0.05$ compared to talazoparib; linear mixed model analysis with a test of simple main effects, exact $P$ values listed in Supplementary Data 8), **d** tumor volume (mm$^3$) for individual animals; and, **e** fold change in tumor volume from day 0 to end of treatment (mean ± SEM; PDX 167.2M Cx—[a]$P = 0.0063$ compared to vehicle, [b]$P = 0.0181$ compared to talazoparib; PDX 224R-Cx—[a,d]$P < 0.0001$ compared to vehicle, [b]$P = 0.0012$ compared to vehicle, [c]$P = 0.0047$ compared to talazoparib and [e]$P < 0.0001$ compared to carboplatin; PDX 426M-Cx—[a]$P = 0.0196$ compared to vehicle, [b]$P = 0.0013$ compared to vehicle and [c]$P = 0.0358$ compared to talazoparib; one-way ANOVA with post hoc Tukey's test). Source data are provided as a Source Data file.

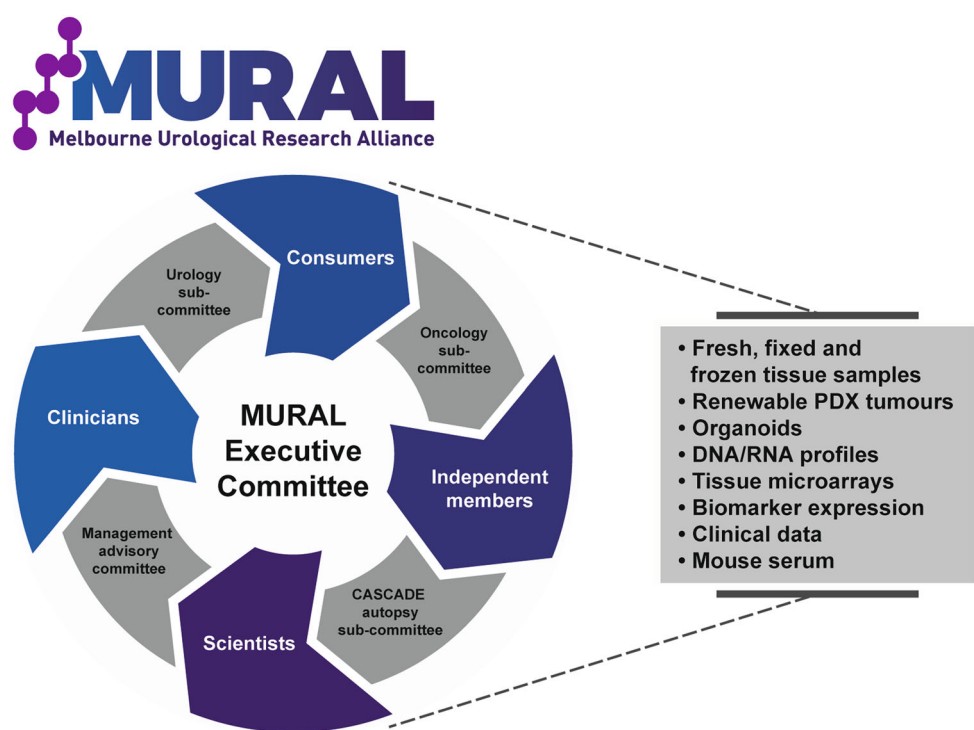

**Fig. 6 Sharing and distribution of the MURAL PDX collection.** MURAL is a PDX biospecimen resource that is available for collaborative, investigator-led research. This schematic shows the organisation of MURAL, with four working sub-committees that are overseen by an executive committee. Resources available from MURAL include PDX tumors, tissue samples, tissue microarrays (TMAs), organoids, DNA/RNA profiles, and clinical data.

substantially increasing the number of PDXs available for prostate cancer research. The PDXs also complement other preclinical models of prostate cancer, including organoids, conditionally reprogrammed cells, and explants, each with their own benefits and limitations[26,37,47–50].

A strength of our PDX cohort is the ability to study particular subtypes of prostate cancer, because the PDXs represent diverse clinical trajectories, pathologies, and genomic features. Our cohort includes 15 PDXs (24% of collection) from treatment-naïve tumors, and the remainder are of castrate-resistant disease post-treatment, including treatment with abiraterone, enzalutamide, lutetium PSMA, and the PARP inhibitor niraparib. Notably, tumors collected from individual metastases or different stages of disease from the same patient provide opportunities to study

intra-patient heterogeneity and disease progression. For example, we recently reported that Patient 167 acquired an *AR* amplification in the progression from primary castrate-sensitive disease to metastatic CRPC[51]. A limitation was the low take rate of bone metastases, a common site of prostate cancer metastasis. This is likely due to it being more difficult to prepare bone than soft tissues for sub-renal grafting.

Multiple mechanisms of castration-resistance were identified, including structural rearrangements, amplification, and mutations of the *AR*, as well as AR-null and neuroendocrine phenotypes. Many genetic drivers of prostate cancer were also prominent, such as mutations and deletions of *TP53* and *PTEN*, *MYC* amplifications and alterations in DNA damage repair pathways, including germline and somatic *BRCA2* loss. Therefore,

the MURAL PDX cohort provides depth to the existing PDX collections, greatly expanding the potential for preclinical testing and understanding mechanisms of therapy resistance.

Internationally, more work is required to share PDXs between laboratories for coordinated research efforts. To facilitate the distribution of MURAL resources, we established an overarching consortium to provide governance and oversight. MURAL consists of an executive committee, a management advisory committee, and sub-committees focussing on specific subsets of tumors, such as autopsy specimens collected through the CASCADE rapid autopsy program[35]. The MURAL membership includes scientists, clinicians, patient advocates, independent representatives, and specialists in statistics and bioinformatics who recommend resource allocation. This maintains the active engagement of clinicians and patient advocates in the decision-making process of sharing resources with the research community and pharmaceutical industry.

PDXs are serially passaged in vivo for years, so misidentification and cross-contamination are possible, particularly as PDXs are increasingly shared between laboratories. To ensure the fidelity of PDXs, we perform routine pathology, where tissue morphology and biomarker expression are assessed every PDX generation; regular RNA and DNA sequencing; and authentication using STR profiling[52]. Another major issue has been the development of mouse or human lymphoma in PDXs[19], which is monitored by routine immunohistochemistry for CD45+. Overall, our data show that MURAL PDXs are stable over long-term propagation and their pathological features are maintained. Cryopreservation protocols have also been optimized for MURAL PDXs[53], so it is possible to restore earlier generation PDXs and distribute them to other researchers. Thus, the MURAL cohort provides authenticated PDXs that can readily be shared.

Based on the spectrum of pathologies and genetic alterations among MURAL PDXs, appropriate models can be selected for preclinical testing, either grown as in vitro organoids or in vivo tumors. Herein, we exemplified this using the $1 \times 1 \times 1$ design, identifying talazoparib plus carboplatin as a promising combination therapy across the research-ready PDXs. In subsequent expansion studies, 4/5 PDXs responded to the combination therapy despite no deleterious alterations identified in DNA damage repair genes, with the exception of heterozygous loss of BRCA2 in PDX-305R. Thus, the MURAL PDX resource is suitable for testing promising therapeutic strategies in primary and metastatic prostate tumors.

The PDXs treated with talazoparib and carboplatin were established from patients who had not received these treatments in the clinic, so we cannot compare PDX versus clinical responses for these agents. Other studies have shown high concordance between clinical outcome and PDX responses for several solid tumors, including prostate, breast, and ovarian cancer[51,54,55]. However, due to intra-patient heterogeneity, variation between metastases is to be expected in some cases. For example, in the MURAL collection, PDX 201.1A and :201.2A were established from different metastatic sites from the same patient; however, they have substantial phenotypic and genetic heterogeneity, and different responses to PARP inhibitor combination therapies. Therefore, it is important to re-examine the sensitivity of each tumor in vivo even if the patient's clinical response to the therapeutic agent is known.

The MURAL PDX collection delivers robust preclinical models that recapitulate the spectrum of clinical, genomic, and phenotypic heterogeneity observed in the clinic may increase the ability to identify putative biomarkers of response ahead of clinical development and significantly improve the success of clinical drug development. Although the major genetic alterations and phenotypes found in patient tumors are represented by MURAL PDXs, the purpose of this collection is not for personalized medicine, but rather to identify promising therapies that can be prioritized for early phase clinical trials. Proof-of-concept preclinical studies using this resource and other PDX collections will provide greater confidence in moving compounds into the next phase for clinical evaluation.

## Methods

**Patient samples.** In this study, 208 prostate cancer specimens were collected from 88 patients obtained between 2012 and 2020. Specimens were obtained from multiple sources: (1) localized, treatment-naive tumors obtained at radical prostatectomy ($n = 46$); and CRPC specimens collected during (2) TURP ($n = 9$), (3) metastatic biopsy ($n = 14$), (4) surgical resection of symptomatic metastases ($n = 5$) or (5) rapid autopsy ($n = 134$) (according to the CASCADE collection process detailed in Alsop et al.[35]). All specimens were obtained with informed, written consent (Human Ethics Approvals Monash Health RES-19-0000-407E at Epworth Eastern Hospital, Monash Health RES-20-0000-107C at Cabrini Hospital, 1636 at Monash University, 15/98, 97_27 and 18/76 at Peter MacCallum Cancer Centre and E55/1213 at Eastern Health).

Clinicopathological features of each patient specimen are presented in Supplementary Data 1. Tumor regions within prostatectomy specimens were identified by uro-pathologists (TissuPath Specialist Pathologists, Mount Waverly, Australia). Tissue viability and tumor content were assessed by haematoxylin and eosin staining[56]. Tumor samples were transported to Monash University laboratories on ice in transport medium (Roswell Park Memorial Institute (RPMI) 1640 medium (Life Technologies, California, USA) supplemented with 10% (vol/vol) fetal calf serum (FCS; Thermo Scientific, HyClone, Massachusetts, USA), 1% (vol/vol) penicillin-streptomycin (Life Technologies), 0.5 μg/ml amphotericin B antimycin (Life Technologies) and 100 μg/ml gentamicin (Life Technologies)). Prostate samples were processed within 2 h following surgery and routine pathology for clinical purposes was not impeded at any time.

**Animals.** Host recipients of PDX tissue were 6–8-week-old male non-obese diabetic-severe combined immunodeficiency (NOD-SCID) or NOD-SCID interleukin 2-receptor gamma chain knockout (NSG) mice. NOD-SCID mice were obtained from the Animal Resources Centre (Canning Vale, Australia). NSG mice were purpose bred at Monash Animal Research Laboratories (Monash University, Monash Breeding Colony approval number MMCA 209/25BC and 15160). All animal handling and procedures were approved by the Monash University Standing Committee of Ethics in Animal Experimentation (SOBSA/A/2010/67, MARP/2014/085, MARP/2014/119, MARP/2016/016, 17963, 17086 and 22185). All mice were bred and housed under controlled temperature (22 °C) and lighting (12:12 h light-dark cycle), and were fed a chow diet ad libitum.

**Patient-derived xenografts.** PDXs were established by the Monash Urological Research Alliance (MURAL) according to human and animal ethics approvals[21]. Patient specimens were manually dissected using a sterile scalpel blade into 4 mm³ pieces and stored in a transport medium at 4 °C until xenograft preparation. Approximately 20% of tumor pieces were fixed in 10% (vol/vol) formalin neutral buffered solution (Sigma Aldrich) for primary specimen histology. All specimens were implanted under the renal capsule of male mice for a period of up to 52 weeks. A maximum of six grafts were implanted into each host mouse (3 per kidney). At the time of surgery, host mice were supplemented with a subcutaneous 5 mm silastic testosterone implant to increase circulating testosterone levels[56–59].

**Serially transplantable xenograft tissues.** To establish serially transplantable PDXs, we strategically selected a broad spectrum of aggressive prostate cancer tumors from men with high-risk or aggressive prostate cancer. This included samples derived from radical prostatectomy or TURP from men with a Gleason Score ≥8 (ISUP 4–5), or were determined to have high-risk features, including strong family history or the presence of IDCP, ductal or neuroendocrine pathology at the time of biopsy. In addition, we collected metastatic samples at the time of biopsy, surgery or rapid autopsy[35].

At the time of engraftment, a 5 mm testosterone pellet was implanted subcutaneously to supplement host testosterone levels. All grafts were established and maintained in testosterone-supplemented animals for a period of up to 52 weeks. Mice were monitored bi-weekly for signs of tumor growth. At the time of tissue harvest, blood was collected via cardiac puncture and mice were euthanized. Grafted tissues were dissected from the kidneys and measured with callipers. If grafts were observed to increase in volume by >10x, tissues were manually dissected using a sterile scalpel blade into 4 mm³ pieces and re-implanted into the sub-renal grafting site of host mice supplemented with testosterone. The remaining tissue pieces were fixed in 10% (vol/vol) formalin neutral buffered solution, stored in RNAlater (Sigma Aldrich), snap frozen in liquid nitrogen and/or cryopreserved in FCS with 10% DMSO and 5 μM Y-27632[53].

PDXs were considered to be serially transplantable when they reached three generations of passaging into new host mice, with tumor volume increasing by at least 10-fold per generation. Once serially transplantable PDXs were established, grafts were transferred to the sub-cutaneous site of host mice to allow for external

monitoring of tumor size using callipers. Over time, sub-cutaneous grafts were then transferred to castrate host mice to establish castrate-resistant PDXs. To do this, grafts were established in testosterone-supplemented host mice until tumor volume reached 200 mm³, at which point host mice were castrated and the testosterone pellet removed. Once tumors reached 1000 mm³, they were re-grafted directly into castrated animals. This occurred between generations 4–31 for each PDX (Supplementary Data 1).

**Authenticity of PDXs.** The identity of each PDX was periodically authenticated by profiling STRs with the GenePrint 10 System (Promega) at the Australian Genome Research Facility, Melbourne. Germline DNA or early generation PDXs were used as controls. PDXs passed authentication when there was ≥80% match in alleles for *amelogenin, CSF1PO, D13S317, D16S539, D5S818, D7S820, TH01, TPOX,* and *vWA*, consistent with standards for human cell lines.

PDXs were also screened for lymphoma using immunohistochemistry, with a combination of markers to confirm the phenotype of human prostate epithelial cells (human CK8/18, human mitochondria, PSA, PSMA, AR) and exclude the presence of lymphoma (CD45). Antibodies are listed in Supplementary Data 9. Any PDXs that were contaminated with lymphoma were no longer regrafted.

**Histology.** To account for heterogeneity of prostate cancer tissue, patient specimens and first-generation PDXs were sectioned into 5 μm serial sections, with tissue pathology analyzed in every 20th section across the tumor tissue. For serially transplantable PDXs and organoids, we selected representative sections of each sample. All tissues were stained using hematoxylin and eosin (H&E) for pathological assessment. Immunohistochemistry was performed using the Leica BOND-MAX™ automated system (Leica Microsystems, Mount Waverley, Australia; Supplementary Data 9). Immunohistochemistry is performed on every fifth generation of PDX tissue.

**Analysis of tumor take rate.** To determine the amount of tumor within primary prostate specimens, each tissue was stained using double immunohistochemistry for p63 and AMACR expression. Hematoxylin and eosin staining on primary tumors was also assessed by independent trained pathologists Associate Professor John Pedersen and Dr David Clouston (Supplementary Fig. 7). To determine tumor take rates within PDX tissues, all specimens were assessed by hematoxylin and eosin staining and double immunohistochemistry for CK8/18 and p63 to detect regions of epithelial growth. Sections containing CK8/18⁺ epithelia lacking p63⁺ basal cells were also stained using immunohistochemistry for AMACR to determine if this protein was expressed. The tumor take rate for each PDX was determined by the presence of AMACR⁺/CK8/18⁺/p63⁻ epithelial cells (Supplementary Data 2; Supplementary Fig. 7); however, rare AMACR⁻/CK8/18⁺/p63⁻ tumor regions were assessed by multiple trained investigators to ensure correct identification. PDX tissues containing only CK8/18⁺/p63⁺ glands were considered to be benign (Supplementary Fig. 7). The tumor take rate was not dependent on the size or number of tumor regions.

**Biomarker quantification.** Ki67 and cleaved caspase 3 stainings were performed on three cancer-containing sections for each primary specimen and PDX graft containing tumor regions. Positive cells within tumor regions were determined using ImageScope analysis software (Aperio). Staining intensity for all other markers was analyzed independently by at least two researchers and given a score of 0, 1+, 2+, 3+.

**DNA and RNA analysis of PDXs**

*Targeted DNA sequencing.* All sequencing data are available through the NCBI Sequence Read Archive, under BioProject PRJNA675382. Two gene panels were used for targeting DNA sequencing of distinct PDX samples. The first platform, denoted as the Garvan panel (Supplementary Data 4), is a cancer gene sequencing panel (PV2) designed through Roche/Nimblegen spanning 633 cancer-associated genes (2.01 Mb target region)[60]. The second platform, denoted as the Twist panel (Supplementary Data 4), is a custom gene capture panel from Twist Biosciences targeting a total of 2.13 Mb of exonic sequence from 16,875 regions in 662 genes. It includes all genes from the Garvan panel, and 29 additional genes of interest. Targeted sequencing regions for both panels are included in Supplementary Data 4. Hybridization capture and library preparation were performed in batches of 8 according to the manufacturer's instructions with the exception of fragmentation time, which was reduced to 16 min. Libraries were barcoded and sequenced on the Illumina NextSeq platform using 75 bp paired-end reads.

*Alignment and variant calling.* Raw reads were trimmed using cutadapt[61] (v2.1; parameters -q 15 -m 50) and aligned to hg19 (GRCh37) using BWA-MEM[62] (v0.7.17). Picard[63] (v2.17.3) was used to sort bam files and mark duplicates. The GATK[64,65] (v3.8) best practices workflow was followed for variant calling. Matched germline DNA was not available for all patients, so variant calls for PDXs were identified using Haplotype caller with default settings.

*Variant curation.* Ensembl Variant Effect Predictor (VEP)[66] v90 was used to annotate results to GRCh37. Silent/synonymous mutations were excluded and we only examined results from the canonical transcript. Reads with <10 reads on the alternate allele were excluded. Variants with high IMPACT scores from VEP (i.e., stop gain, frameshift, indels in exons) were retained. Variants with gnomAD frequency >0.001 were excluded. Variants classified as moderate impact by VEP (i.e., missense mutations or in-frame frameshifts) were included if they fulfilled one of four criteria: (1) CADD_PHRED score of >30, (2) 'Pathogenic' or 'Likely Pathogenic' in Clinvar, (3) deleterious in dbNSFP meta scoring MetaSVM ('D') or MetaLR ('D'), or (4) pathogenic/deleterious in FATHMM-mkl ('D') and SIFT (deleterious) and PolyPhen (damaging or probably damaging). Variants above allelic frequency 0.75 passing the above criteria were manually reviewed, and then excluded if they had conflicting pathogenicity (i.e., marked benign in Clinvar despite pathogenic prediction) or were suspected as germline if reported in multiple PDXs, had a dbSNP ID and/or found in gnomAD.

*Copy number analysis.* CNVkit[67] (v0.9.3) was used to call copy number changes with normal samples pooled into a single reference for each sequencing platform. Copy number was called using batch mode with the drop-low-coverage option against the GRCh37.73 reference. Copy numbers were reported in log2 scale. Thresholds for calling copy number changes for homozygous, hemizygous loss, single copy gain, and hyper amplified were log2(0.2/2), log2(1.2/2), log2(2.8/2), and log2(8/2). Whole copy number values were calculated relative to ploidy of 2. Cut-offs were intended to be a conservative approach to identify whole copy number gain or loss, with an additional threshold of 0.2 to account for heterogeneity within the tumor.

Copy number calls from cBioPortal[68,69] resource Prostate Adenocarcinoma[24] consisting of recurrent amplifications or homozygous deletions in >1% of cases that were also annotated as pathogenic in OncoKB formed the basis for a curated subset set of copy number changes. Copy number changes in the PDX cohort that are found in this set, and that don't conflict with the known direction of effect (i.e., amplifications were not included if the gene has only had reported deletions) were included in our curated set of copy number alterations. In figures using copy number information, copy number alterations are only included if they are found in our curated set of prostate cancer-associated genes previously reported in patient cohorts[24,25] (Supplementary Data 4).

*Percent genome alteration.* Percent genome alteration (PGA) was calculated using segmented copy number calls from cnvkit including off-target reads, estimated as ratio of the number of base pairs spanning regions with copy number changes to the total number of base pairs without changes. Segments with copy number change ±0.5 were used to determine which regions of the genome were altered.

**3′ Transcriptomics of PDXs.** Lexogen 3′ Quantseq FWD + Illumina HiSeq 2000/2500 was used to sequence 89 samples from 39 PDXs (distinct samples from different PDX generations). This spanned a combination of early generation lines (reference) and the most recent generation at the time of sequencing (working). We assigned a representative sample for each line. Where possible this was from the most recent working line in a castrate host, otherwise it was from the most recent sample maintained in a testosterone-supplemented host. Representative samples are opaque in Fig. 3a, while other samples sequenced from the same line are transparent. The full list of samples sequenced is listed in Supplementary Data 10. Reads were reviewed using FastQC to ensure they passed quality control. Reads were trimmed using cutadapt 1.7.1 [with arguments -a AAAAAAAAAA -q 10 -m 20 -u 12]. Trimmed reads were aligned to hg38 using hisat (2.0.4) with default settings, and mouse reads aligned to mm10. Xenomapper (1.0.1) was used to select only primary specific human reads. Rsubread (1.28.1) function featureCounts was used to generate counts matrix from the sam files. EdgeR 3.28.1 used to transform reads into counts per million. Genes were included if they had a minimum of 1 CPM reads across at least 10 samples. As the lexogen QuantSeq FWD only produces one fragment per transcript, normalizing for read length was not required.

**Single-cell RNA transcriptome analysis**

*Dissociation of PDXs.* PDXs were harvested from host mice and cut into 2 × 2 mm pieces using a scalpel. Tumor pieces were digested in RPMI-1640, containing 0.65 U/mL Liberase TM (Roche) and 0.2 mg/ml DNase I (Roche), for 1 h at 37 °C, following by lysis of red blood cells using Red cell Lysis buffer (Sigma) for 1 min. Cells were then resuspended in PBS, 1 mM CaCl₂, with 2% FBS and underwent negative selection for viable cells using the Easy Sep Dead Cell Removal kit (Miltenyi), according to the manufacturer's protocol. Viable cells were passed through a 30 μM cell strainer (Miltenyi) to remove cell clumps, and then counted using Trypan blue. Samples with cell viability >80% were resuspended in PBS with 2% BSA and proceeded to single-cell analysis.

*Single-cell RNA-sequencing.* scRNA-Seq for dissociated PDXs was performed using the 10X Genomics Chromium Single Cell 3′ Library & Gel bead Kit V3.0, according to the manufacturers protocol (CG000183 Rev C). Briefly, ~5000 PDX cells were used as input per sample. Cell encapsulation in microfluidic droplets yielded ~4000 recovered single-cell transcriptomes per sample. After reverse transcription,

barcoded-cDNA was purified using SILANE Dynabeads followed by 11 cycles of PCR-amplification. SPRIselect purification was performed on an Agilent Bioanalyzer High Sensitivity chip to quantitate the fragment size and concentration of the amplified cDNA.

Libraries were sequenced on an Illumina NovaSeq6000 with 151 bp paired end reads. Sample profiling in CellRanger v3.0.2 indicated sequencing yielded 105,281,610 million reads for sample PDX 224R and 195,877,712 million reads for sample PDX 287R, with sequencing saturation of 18% for PDX 224R and 31% for PDX 287R, barcode Q30 scores of 95.5 and 95.6% for those two samples, respectively (Supplementary Data 11).

*Single-cell RNA-Seq analysis.* XenoCell v 1.0 was used to align transcripts to the GRCh38 human reference genome and mm10 mouse genome[70], human-specific cellular barcodes were retrieved containing a max of 10% of host-specific reads. Extracted tumor cells were then processed using Alevin tool (Salmon Software v1.2.1) to obtain unique molecular identifiers (UMIs; Supplementary Data 11) and generate a cell by gene count matrix[71], which was imported into Seurat (v 3.2.0) for downstream analysis[72].

We excluded outlier cells that expressed fewer than 1000 genes and had unusually gene count, transcript count, and mitochondrial gene fraction, according to sample-specific thresholds (Supplementary Data 11). We also excluded genes expressed in fewer than 50 cells. The SCTransform function was then used to log-normalize and scale counts to 10,000 transcripts per cell and detect highly variable features. Principal component analysis (PCA) was performed using the top 2000 most highly variable features and used to cluster cells by Uniform Manifold Approximation and Projection (UMAP). Cell clusters were identified using Seurat FindClusters function. The ClusterTree R package was used to determine the optimal resolution and number of clusters for each sample[73]. Expression of selected genes were plotted as Log2TP10K. Expression of selected gene signatures were calculated per-cell using the AddModuleScore function from Seurat.

**Organoids from PDX tissues.** Freshly harvested PDX tissue was trimmed of excess connective tissue, finely minced with scalpels and digested for 40 min at 37 °C in RPMI-1640 containing 0.65 U/ml Liberase TM (Roche) and 0.2 mg/ml DNase I (Roche). Digested samples were centrifuged, filtered through 100 μm cell strainers, treated with Red Blood Cell Lysis Buffer (Sigma) and washed with RPMI containing 10% FBS, 10 U/mL penicillin and 10 mg/mL streptomycin. Cells were filtered again using a 70 μm cell strainer, centrifuged, and counted. At this stage, cells were either seeded or cryopreserved ($5 \times 10^6$ cells in 90% FBS, 10% DMSO with 10 μM Y-27632 dihydrochloride (Selleckchem)). To generate organoids, viable cells (fresh or thawed from cryopreserved stocks) were seeded in 100% growth factor reduced, phenol red-free, IdEV-free Matrigel (Corning). Typically, $1–1.5 \times 10^5$ cells were seeded in 25–40 μl of Matrigel in 24 or 48-well plates. Cells were cultured in ENR media or ENR-2 media[27,74]. ENR media was advanced DMEM/F-12 media (GIBCO) containing 1% penicillin-streptomycin, 2 mM Glutamax, 1 nM DHT (Sigma), 1.25 mM N-acetylcysteine (Sigma), 50 ng/ml EGF(Sigma), 10% Noggin conditioned media, 10% R-spondin 1 conditioned media (Monash BDI Organoid Program), 500 nM A83-01 (Sigma), 10 ng/ml FGF10 (PeproTech), 5 ng/ml FGF2 (PeproTech), 10 mM Nicotinamide (Sigma), 10 μM SB202190 (Sigma), 2% B27 (GIBCO), 1 μM Prostaglandin $E_2$ (Tocris). ENR-2 media had the same formulation, except without the addition of 1.25 mM N-acetylcysteine and 10 μM SB202190 (Sigma). Y-27632 dihydrochloride (10 μM) was also added to the culture medium during organoid establishment, but not for subsequent passages. DHT (1 nM) was added to organoid media for cells from PDXs grown in testosterone-supplemented mice, but omitted from the media for cells from PDXs grown in castrated mice. For histopathology, organoids were fixed in 10% formalin for at least 1 h, and then embedded in paraffin. Immunohistochemistry was performed using antibodies in Supplementary Data 9 and scored using Aperio ImageScope software as per the PDX tissue.

**Organoid growth assay.** Digested PDX cells were seeded in Matrigel and grown until organoids reached ~50 μm in diameter or until confluency. Organoids were dispersed into single cells with TrypLE Express (GIBCO), counted, and replated over subsequent passages. Cultures were terminated if cell viability dropped below 25%. For actively growing cultures, we calculated the population doubling level as described by the American Type Culture Collection. Briefly, the population doubling level (PDL) = 3.32 (log Xe – log Xb) + S, where Xb was the number of cells seeded at the beginning of the passage, Xe was the number of cells counted at the end of the passage, and S was the starting population doubling from the previous passage.

**RNA sequencing and analysis of PDXs and organoids.** RNA was isolated from matching PDXs and organoids (Passage 3) using RNeasy Micro kit (Qiagen) as per the manufacturer's instructions. RNA sequencing was performed according to the 3′ Transcriptomics described above. Lowly expressed genes were first removed by retaining genes that have at least 1 count per million (CPM) in at least two samples. Library normalization was then performed by calculating the CPM, followed by a log2-transformation with a pseudocount of 1 (i.e., log2(CPM+1) transformation). To visualize the relationship between the transcriptomes, principal component analysis (PCA) was performed using all genes. Subsequently, for visualizing the

expression patterns, the CPM expression for biological duplicates for each condition were averaged and scaled for plotting onto a heatmap. Scatter plots of corresponding PDX and organoids samples were also plotted and the Pearson correlation was calculated to assess the similarity of the transcriptomes. Single-sample gene set enrichment analysis was used to calculate the enrichment of MSigDB50 pathways and the Beltran neuroendocrine signature[22,23,75]. Xenomapper (1.0.1) was used to remove mouse reads.

**PDX clinical trial and expansion studies.** For preclinical drug testing in vivo, we used a one animal per model per treatment approach ($1 \times 1 \times 1$)[1,11], to rapidly screen drug combinations across the PDX cohort. PDXs were established subcutaneously (1 graft/mouse) until tumor volume reached 100 mm³. Mice were systematically assigned to treatment and vehicle groups as their tumors reached the appropriate starting size. Mice were then treated with 0.33 mg/kg talazoparib (5 doses/week by oral gavage, dissolved in PBS with 6% solutol and 10% dimethylacetamide) in combination with one of the following agents: 10 mg/kg apalutamide (5 doses/week by oral gavage, dissolved in sterile water with 2% DMSO, 40% polyethylene glycol 300 and 2% Tween 80), 0.5 mg/ml azacitidine (5 IP doses/week, dissolved in sterile water with 5% DMSO and 30% polyethylene glycol 300), 30 mg/kg AZD-1775 (2 doses/week by oral gavage; dissolved in sterile water with 2% DMSO, 40% polyethylene glycol 300 and 2% Tween 80), 50 mg/kg carboplatin (1 IP dose/week, dissolved in sterile water), 10 mg/kg docetaxel (1 IP dose/week, dissolved in PBS with 5% Tween 80 and 5% ethanol), 10 mg/kg enzalutamide (5 doses/week by oral gavage, dissolved in 1% carboxymethylcellulose with 5% DMSO and 1% Tween 80, followed by sonication); and, 30 mg/kg VX-970 (5 doses/week by oral gavage, dissolved in sterile water with 5% DMSO and 45% polyethylene glycol 300). Mice either received combination treatment or treatment-matched vehicle, with one mouse per treatment per PDX. Mice were treated for 28 days, and tumor volume was measured with callipers three times weekly throughout the treatment period. In some cases, tumor volume reached the maximum ethical limit of 1000 mm³ during the treatment period and tumors were harvested to comply with animal ethics approvals. Based on the growth of our PDXs over 4 weeks, we used pre-defined the thresholds for the responder (<100% growth), partial responder (100–300% growth and ≤50% volume of matched vehicle), and non-responder (>300% growth), using the change in graft volume from the start of treatment.

Based on the results of the $1 \times 1 \times 1$ study, we selected talazoparib + carboplatin for an expansion study in 5 PDXs. Grafts were established subcutaneously (1 graft/mouse) until tumor volume reached 100 mm³, at which point mice were treated with vehicle, 0.33 mg/kg talazoparib, 50 mg/kg carboplatin or combination (6–8 mice/treatment group). Power calculations determined that this sample size provided 80% power to detect 50% change in tumor volume. For randomization, mice were systematically allocated to treatment groups as grafts reached the starting volume. Mice were treated for 28 days, unless tumors reached 1000 mm³ before the end of the treatment period, with calliper measurements performed three times a week to monitor tumor growth. For both the $1 \times 1 \times 1$ study and the expansion study, pieces of each tumor were formalin-fixed, stored in RNA later (ThermoFisher) and frozen in OCT (Sakura). The investigators involved in treatment mice were not blinded to the treatment groups but were blinded to tumor measurement data.

**Bioinformatics software.** Analysis of RNA-seq, targeted DNA sequencing, and single-cell sequencing results conducted in R version 3.6.1, as well as tidyverse[76] (1.3.0).

Analysis of targeted DNA sequencing utilized cutadapt[61] (v2.1), BWA-MEM[62] (v0.7.17), Xenomapper (1.0.1), Picard[63] (v2.17.3), GATK[64,65] (v3.8), VEP[66] v90 with dbNSFP4.0a, CNVkit[67] (v0.9.3). Targeted DNA alterations were visualised using Oncoprint function from ComplexHeatmap (2.2.0)[77]. RNA-seq sequencing and analysis included cutadapt (1.7.1), Xenomapper (1.0.1), Rsubread (1.28.1), EdgeR 3.28.1, with visualisation using heatmap3 (1.1.7; https://cran.r-project.org/web/packages/heatmap3/) and ggplot2.

Software used for single-cell RNA sequencing analysis included Cell Ranger version 3.0.2, Xenocell version 1.0, Salmon (Alevin tool) version 1.2.1, Seurat version 3.2.0, and Clustree version 0.4.3.

**Statistical analysis.** GraphPad Prism 7.0 software (GraphPad Software, California, United States of America) was used for all analyses except linear mixed model analyses, which were performed using SPSS Statistics (Version 25; International Business Machines Corporation, Armonk, New York, United States of America). Analysis of PDX tumor take rate was compared using an unpaired Students *t*-test or one-way analysis of variance (ANOVA) with Tukey's post hoc multiple comparisons test. The take rates of different sources of tissues were compared using a chi-squared test with Fisher's exact approach for post hoc analysis[78]. To determine the effect of treatments on tumor growth, we used linear mixed model analyses, which account for animals dropping out before the end of the experimental period due to graft volume reaching the maximum ethical limit. A random intercept model with one level was used, with identity covariance structure and restricted maximum likelihood estimation. To compare between treatment groups and across time within each treatment group, a test of simple main effects was used, with fixed effects set as treatment, time, and the interaction between treatment and time.

All data are expressed as mean ± standard error of the mean (SEM). All analyses were two-sided and test for normality were performed on all datasets. Statistical significance was set at $p < 0.05$. Experiments were not blinded. R 3.6.3 and prcomp function were used for principal component analysis. Singscore 1.7.5 and GSEAbase 1.48.0 were used to calculate gene set enrichment scores.

**Reporting summary**. Further information on research design is available in the Nature Research Reporting Summary linked to this article.

## Data availability

The targeted DNA sequencing, bulk RNA sequencing, and single-cell RNA sequencing data that support the findings of this study have been deposited in the Sequence Read Archive database (https://www.ncbi.nlm.nih.gov/sra/) under the BioProject accession PRJNA675382. The curated set of genomic alterations was based on data downloaded from the cBioPortal resource Prostate Adenocarcinoma[24,25] [https://www.cbioportal.org/study/summary?id=prad_p1000 and https://www.cbioportal.org/study/summary?id=prad_su2c_2015]. Source data are available as Source Data File. The remaining data are available within the Article, Supplementary Information or available from the reviewers upon request.

To request access to MURAL PDXs and/or biospecimens, researchers should contact Dr. Melissa Papargiris, MURAL Project Manager (melissa.papargiris@monash.edu) to initiate an Expression of Interest. Researchers would need to provide evidence of institutional approval to experiment with human PDX tumors and research would be conducted under the conditions of a Materials Transfer Agreement. Source data are provided with this paper.

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

## Acknowledgements

We wish to acknowledge the Traditional Owners of the lands on which these studies were done. We pay our respects to their Elders, past and present. We also acknowledge the members of the Prostate Cancer Research program, the patients, families, and consumers who support our research. We acknowledge the patient representatives, clinical co-ordinators, scientists, and clinicians, who contribute to the Melbourne Urological Research Alliance (MURAL) and its collection of patient-derived models. We acknowledge kConFab including kConFab research nurses and staff, the heads and staff of the Family Cancer Clinics, and the Clinical Follow-Up Study (which has received funding from the NHMRC, the National Breast Cancer Foundation, Cancer Australia, and the National Institute of Health (USA)) for their contributions to this resource, and the many families who contribute to kConFab. We thank Mark McCabe, Chia-Ling Chan, and Alex Swarbrick for providing access to targeted DNA sequencing technology; Kathryn Alsop, Damien Bolton, Nathan Lawrentschuk, Ross Snow, and Sri Apu for provision of clinical samples; and Melissa Bullock for technical support. This work was supported by the National Health and Medical Research Council, Australia (1102752, 1090204, 1138242, 1185616, 1077799, 1156570, 1140222, 1059855, and 1121057; the US Department of Defense through the Prostate Cancer Research Program (G.P.R. W81XWH1810349; opinions, interpretations, conclusions, and recommendations are those of the authors and are not necessarily endorsed by the Department of Defense); Department of Health and Human Services acting through the Victorian Cancer Agency (MCRF15023, MCRF18017, MCRF18012, MCRF17005, L.F. MCRF16007, CAPTIV Program); Prostate Cancer Foundation of Australia (YI 0417, PIRA1519, NCG1616); Peter MacCallum Cancer Foundation (1752); Monash University (Bridging Fellowship L.P.); the CASS Foundation (7139, 8669, 8575); the EJ Whitten Foundation; Movember Foundation (Global Action Plan 1); Movember and National Breast Cancer Foundation Collaboration Initiative Grant (MNBCF-17-012); Movember and MRFF UpFront PSMA Prostate Cancer Research Alliance (PCRA); the Peter and Lyndy White Foundation; and TissuPath Pathology. This research was supported by the Monash University Histology Platform, Monash University Animal Research Laboratories, Monash Biomedicine Discovery Institute Organoid Program, and the NeC-TAR Research Cloud, a collaborative Australian research platform supported by the National Collaborative Research Infrastructure Strategy.

## Author contributions

G.P.R., A.K.C., L.H.P., M.G.L., and R.A.T. had full access to all the data in the study and take full responsibility for the integrity of the data and the accuracy of the data analysis. Study concept and design: G.P.R., A.K.C., L.H.P., M.G.L., and R.A.T. Acquisition of data: A.K.C., L.H.P., R.T., A.B., N.L.L., S.K., R.Q.U., H.W., M.R., B.N., S.O., L.T., W.J., Z.L., and N.C. Analysis and interpretation of data: G.P.R., A.K.C., L.H.P., R.T., A.B., N.L.L., S.K., R.Q.U., H.W., M.R., B.N., L.T., W.J., Z.L., N.C., J.F.O., D.L.G., M.G.L., and R.A.T. Drafting of the manuscript: G.P.R., A.K.C., L.H.P., M.G.L., and R.A.T. Provision of clinical samples including pathology: D.P., C.J.P., S.S., M.P., J.K., H.M., H.T., L.D., R.H., S.S., L.H., M.I., A.A.A., J.G., J.G., J.K., E.M.K., D.M., D.G.M., J.P., D.C., S.N., and A.R., M.F. Critical revision of the manuscript for important intellectual content: R.T., A.B., N.L.L., S.K., R.Q.U., D.L.G., L.F., and M.F. Statistical analysis: A.K.C., L.H.P., R.T., A.B., N.L.L., S.K., R.Q.U., M.G.L., and R.A.T. Obtaining funding: G.P.R., L.F., D.L.G., R.T., M.G.L., and R.A.T. Supervision: D.G.L., R.T., G.P.R, M.G.L., R.A.T. G.P.R., M.G.L and R.A.T. initiated and led the study and wrote the manuscript. All authors were involved in discussions about study design, contributed to the writing, and reviewed the manuscript.

## Competing interests

G.P.R., R Taylor and M.G.L. (Research collaborations: Pfizer, Astellas, Zenith Epigenetics; non-financial); A.A.A. (Speakers Bureau: Astellas, Janssen, Novartis, Amgen, Ipsen, Bristol Myers Squibb; Merck Serono, Bayer; Honoraria: Astellas, Novartis, Sanofi, AstraZeneca, Tolmar, Telix, Merck Serono, Janssen, Bristol Myers Squibb, Ipsen, Bayer, Pfizer, Amgen, Noxopharm, Merck Sharpe Dome; Scientific Advisory Board: Astellas, Novartis, Sanofi, AstraZeneca, Tolmar, Pfizer, Telix, Merck Serono, Janssen, Bristol Myers Squibb, Ipsen, Bayer, Merck Sharpe Dome, Amgen, Noxopharm; Travel + Accommodation: Astellas, Merck Serono, Amgen, Novartis, Janssen, Tolmar, Pfizer; Investigator Research Funding: Astellas, Merck

Serono, AstraZeneca; Institutional Research Funding: Bristol Myers Squibb, AstraZeneca, Aptevo Therapeutics, Glaxo Smith Kline, Pfizer, MedImmune, Astellas, SYNthorx, Bionomics, Sanofi Aventis, Novartis, Ipsen); D.P. (Honoraria: Astellas, Ipsen, Novartis; Advisory/Consulting: Pfizer, Merck Sharp & Dhome, Bristol-Myers Squibb; Research Funding to institution: Medivation, Bristol-Myers Squibb, Merck Sharp & Dhome, Exelixis, Pfizer, Astellas, Bayer, SymVivo, Amgen; Travel: Bristol-Myers Squibb, Pfizer, Amgen); C.J.P. (Honoraria: Advanced Accelerator Applications, Astellas, AstraZeneza, Janssen, Pfizer; Travel support: Bayer, Janssen); S.S. (Grant funding to institution: AstraZeneca, Novartis, Amgen, Genentech, Merck Sharp and Dohme, Merck Serono; Honoraria: Merck Sharp and Dohme, Astra Zeneca and Bristol Myer Squibb); E.M.K. (Consulting or Advisory Role: Astellas Pharma, Janssen, Ipsen; Travel, Accommodations, Expenses: Astellas Pharma, Pfizer, Ipsen, Roche; Honoraria: Janssen, Ipsen, Astellas Pharma, Research Review; Research Funding: Astellas Pharma (institutional), AstraZeneca (institutional)); J Grummet (Honoraria: BK Ultrasound, Biobot, Mundipharma; Travel: Astellas; Owner of MRI PRO Pty Ltd., an online training platform)); L.F. (Grant funding to institution: Pimera). All other authors declare no competing interests.

## Additional information

[1]Prostate Cancer Research Group, Monash Biomedicine Discovery Institute, Cancer Program, Department of Anatomy and Developmental Biology, Monash University, Clayton, VIC, Australia. [2]Cancer Research Division, Peter MacCallum Cancer Centre, Melbourne, VIC, Australia. [3]Sir Peter MacCallum Department of Oncology, The University of Melbourne, Parkville, VIC, Australia. [4]Computational Cancer Biology Program, Peter MacCallum Cancer Centre, Melbourne, VIC, Australia. [5]Department of Medicine, School of Clinical Sciences, Monash University, Clayton, VIC, Australia. [6]Department of Medical Oncology, Monash Health, Clayton, VIC, Australia. [7]Eastern Health and Monash University Eastern Health Clinical School, Box Hill, VIC, Australia. [8]Sheffield Teaching Hospitals NHS Foundation Trust, Sheffield, England. [9]Department of Medical Oncology, Peter MacCallum Cancer Centre, Melbourne, VIC, Australia. [10]Cancer Tissue Collection After Death (CASCADE) Program, Melbourne, VIC, Australia. [11]Australian Prostate Cancer Bioresource, VIC Node, Monash University, Clayton, VIC, Australia. [12]Prostate Cancer Research Group, Monash Biomedicine Discovery Institute, Cancer Program, Department of Physiology, Monash University, Clayton, VIC, Australia. [13]Program in Cardiovascular and Metabolic Disorders, Duke-National University of Singapore Medical School, Singapore, Singapore. [14]Center for Molecular Imaging, Peter MacCallum Cancer Center, Melbourne, VIC, Australia. [15]Department of Urology, Austin Hospital, The University of Melbourne, Heidelberg, VIC, Australia. [16]Department of Surgery, Austin Health, The University of Melbourne, Heidelberg, VIC, Australia. [17]Epworth Healthcare, Melbourne, VIC, Australia. [18]Epworth Freemasons, Epworth Health, East Melbourne, VIC, Australia. [19]Department of Surgery, The University of Melbourne, Parkville, VIC, Australia. [20]Department of Medicine, School of Clinical Sciences, Monash University, Clayton, VIC, Australia. [21]Division of Cancer Surgery, Peter MacCallum Cancer Centre, The University of Melbourne, Melbourne, VIC, Australia. [22]Department of Surgery, Central Clinical School, Monash University, Clayton, VIC, Australia. [23]Australian Urology Associates, Melbourne, VIC, Australia. [24]Department of Medicine, Monash Health, Casey Hospital, Berwick, VIC, Australia. [25]Central Clinical School, Monash University, Clayton, VIC, Australia. [26]The Epworth Prostate Centre, Epworth Hospital, Richmond, VIC, Australia. [27]TissuPath, Mount Waverley, VIC, Australia. [28]Department of Surgery, Monash University, Clayton, VIC, Australia. [29]Department of Urology, Cabrini Institute, Cabrini Health, Melbourne, VIC, Australia. [30]These authors contributed equally: Gail P. Risbridger, Ashlee K. Clark, Laura H. Porter, Mitchell G. Lawrence, Renea A Taylor. ✉email: gail.risbridger@monash.edu; renea.taylor@monash.edu

