## [Peer Review File · Nature Communications]

REVIEWER COMMENTS

Reviewer #1 (Remarks to the Author):

Risbridger et al produce a phenomenal resource with the production and verification of a broad array of human prostate cancer PDXs suitable for pre-clinical testing. This is a major milestone in the prostate cancer field and the manuscript is suitable as is for publication. A rare pleasure to read such a polished study that will highly impact the field for years to come.

Douglas Strand

Reviewer #2 (Remarks to the Author):

The study entitled Discovery through preclinical models of patient-1 derived xenografts in urology: MURAL prostate cancer PDX collection by Risbridger et al. reports on the generation of 59 novel PDXs from Prostate Cancer generated in 8 years period. This great team effort provides the scientific community with a much needed and substantial resource for preclinical tastings. I have read with great pleasure this paper and would like to compliment the authors for the thorough work and well thought experimental plan. Given the descriptive nature of the paper I do not have many comments or suggestions. From my perspective if there is a short coming that I would like to see addressed is from the organoid perspective. The figures show do not give them justice.

Reviewer #3 (Remarks to the Author):

PDX paper review

This paper describes a hugely important piece of work, the establishment of a large (n=59 from 30 patients) "bank" of well-characterised and functional prostate cancer patient-derived xenografts, available to collaborators. This is an eagerly-anticipated, in the field, report on the outcomes and will promote many new collaborations and facilitate many translational studies of new and combination therapies. It has supplemental tables with very comprehensive clinical and molecular details for each PDX, and identifies a "research-ready" set of PDXs covering the spectrum of disease which would be a sound basis for pre-clinical studies. In addition, it will be a valuable methods reference for groups wishing to establish their own PDXs. I am highly supportive of this being published but suggest some changes, none very major, which would increase the accessibility of the paper.

1. P6. The 1x1x1 model needs more explanation.
2. Fig 1 –Several shades of green are used for sample sites, which will be difficult for those with colour recognition impairment. Amending the colour scheme to one less reliant on being able to differentiate shades of green would improve this, as would reordering the legend to the same order they appear in the figure, i.e. prostate, other, lung, brain, lymph...
3. P7 The lines grown in castrated animals: it's implied this was after they were first established in testosterone-supplemented animals, would be good to specify how long after this the Cx sublines were established?

4. P9 why was absence of lymphoma a required test?
5. P9/fig 1 this shows data from the latest generation of each PDX. Presumably this is pretty variable. Can info. on what generation this is be included in the figure?
6. P9. It isn't immediately apparent that "AR-positive PDXs" in text refers to "adenocarcinomas" on figure, etc, make this clear. Also, is this ratio of adenocarcinoma:neuroendocrine:mixed representative of the input population, i.e. was take rate similar across the 3 categories of tumour?
7. P10. It's worth pointing out here that bone is the most common metastatic site in prostate cancer, yet none of the bone mets samples (except 1 in spine but not clear if that is bony?) successfully transplanted, so this represents one aspect in which the PDX series does not mirror the human disease situation.
8. Fig 2. It isn't clear what the green rings in A mean. How many of the 63 and 13, respectively, are prostate – all I think? If so why the need for this ring? In the context of a Venn diagram this looks as though the 63 and 13 are a contained, non-prostate subset.
9. P11 "Tumors with mixed pathology were spread across both clusters (Fig. 3a)" – I would say actually this is underplaying the result, which is that the mixed set appear largely at the border between the 2 other groups. However I might be reading it wrong, as the shading scheme in (a) doesn't match up to the legend in (b) which appears to be meant to cover both a and b – what does lightly shaded (but filled in) mean, as opposed to no fill (Cx) or dark fill?
10. P13 grammatical error "Thus, these genomic features (are) represent(ive) of the genomic spectrum"
11. P13. Supp fig 3c. It is surprising that so few changes are seen in the later generation castrate sublines as compared to the earlier generation testosterone-supplemented versions. When Chang et al famously compared castrate-resistant to androgen-sensitive parental xenografts, the AR was amplified in all cases, for example. What happens if you drill down into AR in this comparison?
12. P14: was ability to grow as an organoid also a feature of the "research ready" PDXs?
13. P16: the reader needs to be introduced to talazoparib, and the rationale for choosing this made clear, at the start of the section "preclinical testing of combination therapies in research ready PDXs". Currently it is only first mentioned (but not described) when discussing fig 5a.
14. P19: "organoids grown from PDX tissue" is a bit ambiguous – are recipients themselves able to grow these as organoids?
15. P20 grammar error/typo "rapidly screening therapies single agent and combination treatments"
16. Finally, It would be ideal to have a website reference for interested collaborators, is this likely to happen?

Reviewer #4 (Remarks to the Author):

The authors have reported on the development and categorization of a tremendous resource developed to facilitate collaborative, investigator-led research initiatives in prostate cancer. The PDXs are extremely well categorized and provide an excellent pathological and genomic spectrum of prostate cancer disease states.

-They should at least mention that 0% of bone met-derived tissues were serially transplantable.

-Not clear why they used CD56 for NE marker in Fig. 3C, but SYN for NE marker in Fig. 3G, and CGA in Fig. 3N.

-- The publication of Centenera MM, Mol Oncol. 2018 Sep;12(9):1608-1622 and the work in PDX hormonal regulation should be cited.

They need to define what they mean by "research-ready" at the first instance of using it in the

results section (p14, line 325)

-I would like some comment of the responses of the PDXs to treatment compared to responses of the patients (probably in the discussion)

Discovery through preclinical models of uro-oncology: MURAL collection of prostate cancer patient-derived xenografts

Response to Reviewers' Comments

Reviewer 1

Comment: Risbridger et al produce a phenomenal resource with the production and verification of a broad array of human prostate cancer PDXs suitable for pre-clinical testing. This is a major milestone in the prostate cancer field and the manuscript is suitable as is for publication. A rare pleasure to read such a polished study that will highly impact the field for years to come.

Response: Thank you for your positive comments and for reviewing our manuscript.

Reviewer 2

The study entitled Discovery through preclinical models of patient-1 derived xenografts in uro-oncology: MURAL prostate cancer PDX collection by Risbridger et al. reports on the generation of 59 novel PDXs from Prostate Cancer generated in 8 years period. This great team effort provides the scientific community with a much needed and substantial resource for preclinical tastings.

I have read with great pleasure this paper and would like to compliment the authors for the thorough work and well thought experimental plan.

Comment: Given the descriptive nature of the paper I do not have many comments or suggestions. From my prospective if there is a short coming that I would like to see addressed is from the organoid perspective. The figures show do not give them justice.

Response: Thank you for your feedback on our manuscript.

To emphasise the organoids, we have added a pie chart to Figure 3 showing the organoid growth of 24 PDXs. This pie chart summarises the information in Supplementary Table 7. We have updated the figure legend for Figure 3 to include this new panel (page 37, lines 927-929) and updated the figure references in the results text. The accompanying results text has also been amended (page 14, lines 325-331). We also updated Supplementary Table 7 to mark the research-ready PDXs, so that it is better integrated with Figure 4.

Reviewer 3

This paper describes a hugely important piece of work, the establishment of a large (n=59 from 30 patients) "bank" of well-characterised and functional prostate cancer patient-derived xenografts, available to collaborators. This is an eagerly-anticipated, in the field, report on the outcomes and will promote many new collaborations and facilitate many translational studies of new and combination therapies. It has supplemental tables with very comprehensive clinical and molecular details for each PDX, and identifies a "research-ready" set of PDXs covering the spectrum of disease

which would be a sound basis for pre-clinical studies. In addition, it will be a valuable methods reference for groups wishing to establish their own PDXs. I am highly supportive of this being published but suggest some changes, none very major, which would increase the accessibility of the paper.

Comment 1: P6. The 1x1x1 model needs more explanation.

Response: We revised the introduction to provide further explanation of the 1x1x1 approach (pages 6-7, lines 156-160), describing it as “an efficient way of screening for active compounds based on striking responses with few biological replicates”.

Comment 2: Fig 1 –Several shades of green are used for sample sites, which will be difficult for those with colour recognition impairment. Amending the colour scheme to one less reliant on being able to differentiate shades of green would improve this, as would reordering the legend to the same order they appear in the figure, i.e. prostate, other, lung, brain, lymph...

Response: We updated the colours in figure 1 so that there is more distinction between different shades. We used a palette suitable for people with colour blindness. We updated subsequent figures to be consistent with this Figure (2, 4, S1).

Comment 3: P7 The lines grown in castrated animals: it's implied this was after they were first established in testosterone-supplemented animals, would be good to specify how long after this the Cx sublines were established?

Response: We revised the Materials and Methods to provide the range of generation numbers for PDXs being transferred to castrated mice (page 25, lines 571-573). We also updated Supplementary Table 1 to clarify the specific generation that each PDX was transferred to castrated mice.

Comment 4: P9 why was absence of lymphoma a required test?

Response: We used CD45 staining to rule out the presence of lymphoma, because it has contaminated PDXs in previous studies, replacing human prostate cancer cells over time. We revised the text (page 9, line 195-197) to note this point and cite previous studies reporting this important issue.

Comment 5: P9/fig 1 this shows data from the latest generation of each PDX. Presumably this is pretty variable. Can info. on what generation this is be included in the figure?

Response: We routinely perform immunohistochemistry to ensure the fidelity of the PDXs, and, in most cases, it is very consistent across generations. The information presented in Figure 1 represents the latest PDX generation, which can be found in Supplementary Table 1. We have clarified this in the figure legend (page 35, line 864) and in the results text (page 9, line 198-199).

Comment 6: P9. It isn't immediately apparent that “AR-positive PDXs” in text refers to “adenocarcinomas” on figure, etc, make this clear. Also, is this ratio of

adenocarcinoma:neuroendocrine:mixed representative of the input population, i.e. was take rate similar across the 3 categories of tumour?

Response: We revised the text to clarify the three main types of PDXs (page 9, lines 200-204). The AR-positive PDXs and adenocarcinomas are all the same, except one adenocarcinoma that lacks AR expression (201.2A/201.2A-Cx), as we've previously reported (Lawrence, 2018, *European Urology*, PMID:30049486).

Regarding the take rates of the three categories of tumors, we only assessed AR staining for this analysis, so the mixed tumors are contained in the AR+ group. We revised the text so this is clear (page 11, line 240).

Comment 7: P10. It's worth pointing out here that bone is the most common metastatic site in prostate cancer, yet none of the bone mets samples (except 1 in spine but not clear if that is bony?) successfully transplanted, so this represents one aspect in which the PDX series does not mirror the human disease situation.

Response: We revised the results text to note the lack of PDXs from bone metastases (page 10, line 233). We also included this as a limitation in the discussion (page 21, lines 493-495). We think this is due to the difficulty in processing patient samples of bone for sub-renal grafting compared to soft tissues.

Comment 8: Fig 2. It isn't clear what the green rings in A mean. How many of the 63 and 13, respectively, are prostate – all I think? If so why the need for this ring? In the context of a Venn diagram this looks as though the 63 and 13 are a contained, non-prostate subset.

Response: The green rings denote that these were all prostate samples, and we included them for continuity with Figure 1. For clarity, we updated the legends within the figure noting what each colour represents.

Comment 9: P11 "Tumors with mixed pathology were spread across both clusters (Fig. 3a)" – I would say actually this is underplaying the result, which is that the mixed set appear largely at the border between the 2 other groups. However I might be reading it wrong, as the shading scheme in (a) doesn't match up to the legend in (b) which appears to be meant to cover both a and b – what does lightly shaded (but filled in) mean, as opposed to no fill (Cx) or dark fill?

Response: We revised the results text to state that the PDXs with mixed pathology were at the border between the two clusters of samples in most cases (page 11, lines 248-249). We also added a legend to indicate that solid shapes denote representative PDX samples and shaded shapes denote replicate PDX samples.

Comment 10: P13 grammatical error "Thus, these genomic features (are) represent(ive) of the genomic spectrum"

Response: We corrected this grammatical error.

Comment 11: P13. Supp fig 3c. It is surprising that so few changes are seen in the later generation castrate sublines as compared to the earlier generation testosterone-supplemented versions. When Chang et al famously compared castrate-resistant to androgen-sensitive parental xenografts, the AR was amplified in all cases, for example. What happens if you drill down into AR in this comparison?

Response: We added a new panel to Extended Data Figure 3 showing *AR* copy numbers in matching PDXs grown in testosterone-supplemented and castrated mice. For two PDXs (394M and 167.2M), there is a trend of increased copies of the *AR* after castration. There was no clear change in *AR* copy numbers or mutations in the other PDXs, although we note that this includes *AR*-null tumours that are unlikely to acquire *AR* alterations after castration. We updated the text to include these points (page 13, lines 310-313). As all of these PDXs were established from samples of castration-resistant prostate cancer, they had pre-existing mechanisms of resistance before being regrafted into castrated mice. We updated the figure legend to include the new panel (Supplementary material page 25) and subsequently updated the figure references to Extended Data Figure 3 throughout the results text.

Comment 12: P14: was ability to grow as an organoid also a feature of the “research ready” PDXs?

Response: No, the ability to grow organoids was not a feature of the “research ready” PDXs. We prioritised PDXs that we could reliably use for *in vivo* preclinical studies and, thus, our criteria focused on how well these PDXs grew *in vivo* (i.e. rapid and consistent turnover time, ability to grow subcutaneously). The fact that some of these PDXs grow as organoids is a bonus. We have amended the results text to clarify our criteria for the research-ready PDXs (page 15, lines 339-341). We also revised Supplementary Table S7 by including a new column showing tumors are research-ready PDXs.

Comment 13: P16: the reader needs to be introduced to talazoparib, and the rationale for choosing this made clear, at the start of the section “preclinical testing of combination therapies in research ready PDXs”. Currently it is only first mentioned (but not described) when discussing fig 5a.

Response: We amended the text to include an introduction to talazaparib at the start of the section “preclinical testing of combination therapies in research ready PDXs” (page 17, lines 392-396).

Comment 14: P19: “organoids grown from PDX tissue” is a bit ambiguous – are recipients themselves able to grow these as organoids?

Response: Yes, cryopreserved organoid cultures are available for distribution to other laboratories and recipients have used them to grow organoids. We revised the text to say “cryopreserved organoid cultures” instead of “organoids grown from PDX tissue” to make this clear (page 20, line 466).

Comment 15: P20 grammar error/typo “rapidly screening therapies single agent and combination treatments”

Response: Thank you, this has been amended.

Comment 16: Finally, It would be ideal to have a website reference for interested collaborators, is this likely to happen?

Response: Yes, we are in the process of developing a website as a reference for collaborators. It will be based on a database that the bioinformaticians in our team have developed as a searchable catalogue of the features of the PDXs. We intend to release this database as a tool for other laboratories who may be interested in tracking their own PDXs over time.

Reviewer 4

The authors have reported on the development and categorization of a tremendous resource developed to facilitate collaborative, investigator-led research initiatives in prostate cancer. The PDXs are extremely well categorized and provide an excellent pathological and genomic spectrum of prostate cancer disease states.

Comment: They should at least mention that 0% of bone met-derived tissues were serially transplantable.

Response: We added this point to the results (page 10, line 233) and discussion (page 22, lines 504-506). We need to optimise the method for implanting bone metastases compared to soft tissues.

Comment: Not clear why they used CD56 for NE marker in Fig. 3C, but SYN for NE marker in Fig. 3G, and CGA in Fig. 3N.

Response: All three markers are used in the clinic or literature to characterise NE pathology. Each PDX has varying levels of CD56, synaptophysin (SYN) and chromogranin A (CGA), so we used the marker that illustrated the expression profile mostly clearly for each PDX. The expression of these markers in each research-ready PDX is summarised in Fig. 4B.

For the single-cell RNAseq data (Fig. 3c, Fig. 3g), 224R expresses all three markers, but as it has highest expression of CD56, we presented this in the UMAP (Fig. 3e). PDX 287R has such low levels of all three markers that the only one we can generate a UMAP for is SYN (Fig. 3i).

For the organoids in Fig. 3n, 305R-Cx expresses all three markers, but the highest levels of CGA.

For clarity, we revised the main text to cite Fig. 4b (which contains NE staining; pages 11 & 12, lines 262 & 271). We also revised the figure legend to note that the highest expressed NE marker is included in each panel (page 38, lines 932-933).

Comment: The publication of Centenera MM, Mol Oncol. 2018 Sep;12(9):1608-1622 and the work in PDX hormonal regulation should be cited.

Response: We included a new sentence in the discussion noting that PDXs complement other patient-derived models of prostate cancer such as explants, as reported by Centenera and colleagues (page 21, lines 493-495).

We also revised the first mention of castrate sublines in the results, to cite several previous studies that examined hormonal regulation of PDXs (page 8, lines 174-176).

Comment: They need to define what they mean by “research-ready” at the first instance of using it in the results section (p14, line 325)

Response: We amended the results text to define research-ready PDXs when they are first introduced, stating that “PDXs are classified as research-ready if they have rapid and consistent turnover in host mice, and are able to grow subcutaneously to allow for continual tumor measurements” (page 15, lines 339-41).

Comment: I would like some comment of the responses of the PDXs to treatment compared to responses of the patients (probably in the discussion)

Response: The PDXs treated with talazoparib and carboplatin were established from patients that had not received these treatments in the clinic, and therefore we did not compare PDX versus clinical response in this manuscript. Previous studies have shown that PDX response often matches clinical outcome, including our recent publication on PDX-167.1R and PDX-167.2M from the MURAL cohort (Porter et al., Journal of Pathology, 2021). However, given intra-patient heterogeneity, we would expect to see variation in the response of PDXs derived from different metastatic sites in some cases. We therefore recommend re-examining the sensitivity of each tumour *in vivo* even if the patient’s clinical response to the therapeutic agent is known.

We have included a new paragraph in the discussion regarding this point (pages 23-24, lines 543-552).

REVIEWERS' COMMENTS

Reviewer #3 (Remarks to the Author):

All comments fully addressed, I look forward to seeing the paper come out

Reviewer #4 (Remarks to the Author):

The authors have done a thorough job of addressing the minor and major critiques of this significant paper.